# An extended catalogue of tandem alternative splice sites in human tissue transcriptomes

Aleksei Mironov[1], Stepan Denisov[1,2], Alexander Gress[3], Olga V. Kalinina[3,4], Dmitri D. Pervouchine[1,5]*

1 Skolkovo Institute for Science and Technology, Moscow, Russia, 2 The Institute for Information Transmission Problems RAS, Moscow, Russia, 3 Helmholtz Institute for Pharmaceutical Research Saarland (HIPS), Helmholtz Centre for Infection Research (HZI), Saarbrücken, Germany, 4 Faculty of Medicine, Saarland University, Homburg, Germany, 5 Faculty of Bioengineering and Bioinformatics, Moscow State University, Moscow, Russia

* d.pervouchine@skoltech.ru

**Data Availability Statement:** The TASS catalogue is available through a track hub for the UCSC Genome Browser https://raw.githubusercontent. com/magmir71/trackhubs/master/TASShub.txt.

## Abstract

Tandem alternative splice sites (TASS) is a special class of alternative splicing events that are characterized by a close tandem arrangement of splice sites. Most TASS lack functional characterization and are believed to arise from splicing noise. Based on the RNA-seq data from the Genotype Tissue Expression project, we present an extended catalogue of TASS in healthy human tissues and analyze their tissue-specific expression. The expression of TASS is usually dominated by one major splice site (maSS), while the expression of minor splice sites (miSS) is at least an order of magnitude lower. Among 46k miSS with sufficient read support, 9k (20%) are significantly expressed above the expected noise level, and among them 2.5k are expressed tissue-specifically. We found significant correlations between tissue-specific expression of RNA-binding proteins (RBP), tissue-specific expression of miSS, and miSS response to RBP inactivation by shRNA. In combination with RBP profiling by eCLIP, this allowed prediction of novel cases of tissue-specific splicing regulation including a miSS in *QKI* mRNA that is likely regulated by *PTBP1*. The analysis of human primary cell transcriptomes suggested that both tissue-specific and cell-type-specific factors contribute to the regulation of miSS expression. More than 20% of tissue-specific miSS affect structured protein regions and may adjust protein-protein interactions or modify the stability of the protein core. The significantly expressed miSS evolve under the same selection pressure as maSS, while other miSS lack signatures of evolutionary selection and conservation. Using mixture models, we estimated that not more than 15% of maSS and not more than 54% of tissue-specific miSS are noisy, while the proportion of noisy splice sites among non-significantly expressed miSS is above 63%.

## Author summary

Pre-mRNA splicing is an important step in the processing of the genomic information during gene expression. During splicing, introns are excised from a gene transcript, and the remaining exons are ligated. Our work concerns one its particular subtype, which

To visualize it, copy and paste the link into the form at http://genome.ucsc.edu/cgi-bin/hgHubConnect#unlistedHubs. ENCODE files that were used in the analysis are available from the https://www.encodeproject.org/ under the accession numbers listed in the Supplementary information. GTEx files used in the analysis are available through GTEx portal https://gtexportal.org/home/ under conditions for General Research Use (phs000424/GRU).

**Funding:** AM and SD were supported by the Skolkovo Institute of Science and Technology Research Grant RF-0000000653 and Russian Foundation for Basic Research grant 18-29-13020-MK (https://www.rfbr.ru/rffi/ru/). AG acknowledges financial support from BMBF grant Sys_CARE (nr. 01ZX1908A) of the Federal German Ministry of Research and Education (https://www.bmbf.de/en/research-funding-1411.html). The funders had no role in study design, data collection and analysis, decision to publish, or preparation of the manuscript.

**Competing interests:** The authors have declared that no competing interests exist.

involves the so-called tandem alternative splice sites, a group of closely located exon borders that are used alternatively. We analyzed RNA-seq measurements of gene expression provided by the Genotype-Tissue Expression (GTEx) project, the largest to-date collection of such measurements in healthy human tissues, and constructed a detailed catalogue of tandem alternative splice sites. Within this catalogue, we characterized patterns of tissue-specific expression, regulation, impact on protein structure, and evolutionary selection acting on tandem alternative splice sites. In a number of genes, we predicted regulatory mechanisms that could be responsible for choosing one of many tandem alternative splice sites. The results of this study provide an invaluable resource for molecular biologists studying alternative splicing.

## Introduction

Alternative splicing (AS) of most mammalian genes gives rise to multiple distinct transcript isoforms that are often regulated between tissues [1–3]. It is widely accepted that among many types of AS, exon skipping is the most frequent subtype [1]. The second most frequent AS type is the alternative choice of donor and acceptor splice sites, the major subtype of which are the so-called tandem alternative splice sites (TASS) that are located only a few nucleotides from each other [4, 5]. About 15–25% of mammalian genes possess TASS, and they occur ubiquitously throughout eukaryotes, in which alternative splicing is common [4]. TASS were experimentally shown to be functionally involved in DNA binding affinity [6], subcellular localization [7], receptor binding specificity [8] and other molecular processes (see [4] for review).

The outcome of the alternative splicing of a non-frameshifting TASS on the amino acid sequence encoded by the transcript is equivalent to that of a short genomic indel. The latter cause broad genetic variation in the human population and impact human traits and diseases [9, 10]. For a different type of alternative splicing with a similar effect on amino acid sequence, alternative microexons, it has been demonstrated that insertion of two amino acids may influence protein-protein interactions in brains of autistic patients [11]. Structural analysis of non-frame-shifting genomic indels revealed that they predominantly adopt coil or disordered conformations [12]. Likewise, non-frame-shifting TASS with significant expression of multiple isoforms are overrepresented in the disordered protein regions and are evolutionarily unfavorable in structured protein regions [13].

The two most studied classes of TASS are the acceptor NAGNAGs [5, 14–16] and the donor GYNNGYs [17]. In these TASS classes, alternative splicing is significantly influenced by the features of the cis-regulatory sequences, but less is known about their function, tissue-specific expression, and regulation [5, 17, 18]. Recent genome-wide studies estimated that at least 43% of NAGNAGs and ∼20% of GYNNGYs are tissue-specific [5, 17]. It is believed that closely located TASS such as NAGNAGs and GYNNGYs originate from the inability of the spliceosome to distinguish between closely located cis-regulatory sequences, and therefore most TASS are attributed to splicing errors or noise [17, 19, 20]. However, it is not evident from the proteomic data what fraction of alternative splicing events and, in particular, of TASS splicing indeed lead to the changes in the protein aminoacid sequence [21–23].

The biggest existing catalogue of TASS, TASSDB2, is based on the evidence of transcript isoforms from expressed sequence tags (EST) [24]. The advances of high-throughput sequencing technology open new possibilities to identify novel TASS [25]. Here, we revisit the catalogue of TASS by analyzing a large compendium of RNA-seq samples from the Genotype

Tissue Expression (GTEx) project [26]. We substantially extend the existing catalogue of TASS, characterize their common genomic features, and systematically describe a large set of TASS that have functional signatures such as evolutionary selection, tissue-specificity, impact on protein structure, and regulation. While it is believed that the expression of TASS primarily originates from splicing noise, here we show that a number of previously unknown TASS may have important physiological functions and estimate the proportion of noisy splicing of TASS. The TASS catalogue is available through a track hub for the UCSC Genome Browser (see Supplementary information).

## Results

### The catalogue of TASS

In order to identify TASS, we combined three sources of data. First, we extracted the annotated splice sites from GENCODE and NCBI RefSeq human transcriptome annotations [27, 28]. This resulted in a list of ∼570k splice sites, which will be referred to as annotated. Next, we identified donor and acceptor splice sites in split read alignments from the compendium of RNA-seq samples from the Genotype Tissue Expression Project (GTEx) [26] by pooling together its 8,548 samples. We applied several filtering steps to control for split read misalignments caused by the presence of germline polymorphisms near splice sites (see Materials and methods for details). This resulted in a list of ∼800k splice sites, which will be referred to as expressed. A splice site may belong to both these categories, i.e. be annotated and expressed, or be annotated and not expressed, or be expressed and not annotated (the latter are referred to as *de novo*). Third, we scanned the transcriptome sequences with SpliceAI software [29] and selected splice sites with SpliceAI score greater than 0.1 excluding splice sites that were previously called expressed or annotated. This resulted in a list of 196,885 sequences that are similar to splice sites, but have no evidence of expression or annotation and will therefore be referred to as cryptic. The combined list from all three sources contained approximately one million unique splice sites (S1(A) Table).

A TASS cluster is defined as a set of at least two splice sites of the same type (either donor or acceptor) such that each two successive splice sites are within 30 nts from each other (Fig 1A). The number of splice sites in a TASS cluster will be referred to as the cluster size. According to this definition, each splice site can belong either to a TASS cluster of size 2 or larger, or be a standalone splice site. Out of approximately one million candidate splice sites in the initial list, ∼177k belong to TASS clusters (S1(B) Table).

About 99% of splice sites in TASS clusters are located in clusters of size 2, 3, 4 and 5 (Fig 1B). In what follows, we confined our analysis to TASS clusters consisting of 5 or fewer splice sites and having at least one expressed splice site (S1(C) Table). This way we obtained ∼151k splice sites; among them ∼69k (46%) expressed annotated splice sites, ∼47k (31%) expressed *de novo* splice sites, ∼5k (3%) annotated splice sites that are not expressed, and ∼30k (20%) cryptic splice sites (Fig 1C). We categorized a TASS cluster as coding if it contained at least one non-terminal boundary of an annotated protein-coding exon, and non-coding otherwise. 87% of annotated TASS and 60% of *de novo* TASS were coding (S1 Fig). This set extends TASSDB2 database, which is limited to TASS separated by 2–12 nt [24], by ∼51k TASS, which are absent from TASSDB2 (Fig 1D). The newfound TASS are less expressed than TASS that are common to TASSDB2; however, the TASS from TASSDB2 that were not identified by our analysis are expressed at a significantly lower level (Fig 1E).

Whereas more than a third of the expressed splice sites are *de novo* (Fig 1C), their split read support is much lower than that of the annotated splice sites (Table 1). We pooled together the read counts from all 8,548 GTEx samples and ranked splice sites within each TASS cluster by

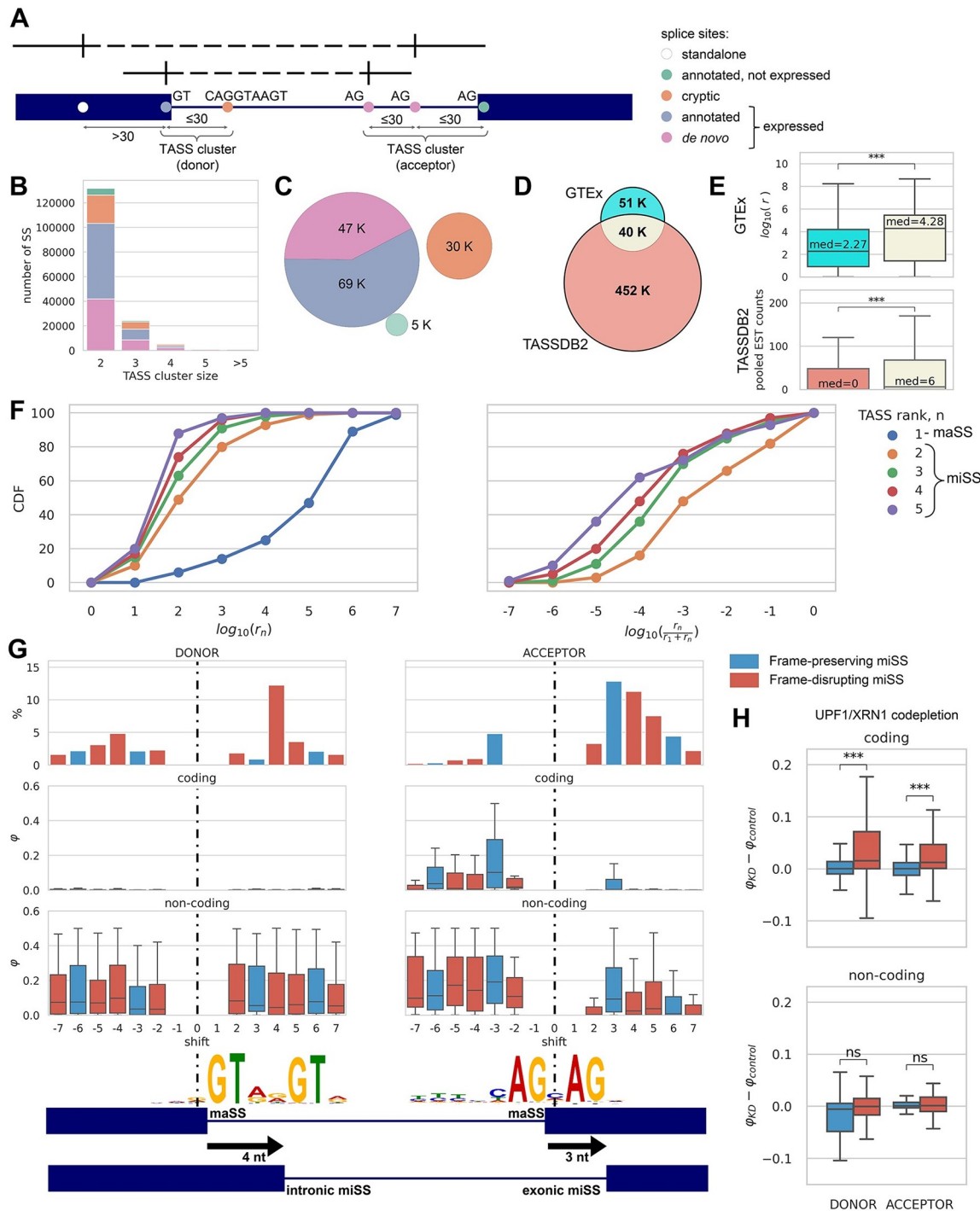

**Fig 1. The catalogue of TASS. (A)** Splice sites are categorized as annotated (GENCODE and Refseq), *de novo* (inferred from RNA-seq) or cryptic (detected by SpliceAI). TASS clusters consist of splice sites of the same type (donor or acceptor) such that each two consecutive ones are within 30 nts from each other. **(B)** The TASS cluster size distribution. **(C)** The number of annotated, de novo and cryptic TASS. Colors are as in panel (A). **(D)** The catalogue of TASS and TASSDB2 database. Only TASS separated by 2-12 nt were counted to match the TASSDB2 content. **(E)** The distributions of the total number of read counts in GTEx (top) and the total number of EST counts provided in TASSDB2 (bottom) for the three TASS categories in panel (D). **(F)** The expression of the major splice sites (maSS, i.e. rank 1) and minor splice sites (miSS, i.e. rank 2 or higher). Left: the cumulative distribution of $r_n$, the number of split reads supporting maSS and miSS. Right: the cumulative distribution of $r_n$ relative to the sum of $r_n$ and $r_1$. **(G)** The distribution of shifts, i.e., the positions of miSS relative to maSS (top) for miSS of rank 2. The relative usage of miSS ($\varphi$) in coding (middle) and non-coding regions (bottom). Logo chart of miSS sequences of +4 donor shifts and of +3 acceptor shifts. Frame-preserving shifts are colored blue and frame-

disrupting shifts are colored red. **(H)** The change of miSS relative usage ($\varphi_{KD} - \varphi_{control}$) upon NMD inactivation for frame-preserving (blue) and frame-disrupting (red) miSS.

**Table 1. Abundance and split read support of annotated and *de novo* TASS.**

|  | expressed TASS | | % of split reads supporting TASS |
| --- | --- | --- | --- |
|  | **number** | **%** |  |
| total | 115,912 | 100% | 100% |
| annotated | 69,330 | 59.81% | 99.83% |
| *de novo* | 46,582 | 40.19% | 0.17% |

**Table 2. The fractions of annotated and *de novo* sites among maSS and miSS.**

|  | *de novo* | annotated | total |
| --- | --- | --- | --- |
| maSS | 9,043 (13%) | 61,130(87%) | 70,173 |
| miSS | 37,539 (82%) | 8,200(18%) | 45,739 |

the number of supporting reads (Fig 1F, left). The dominating splice site (rank 1, also referred to as major splice site, or maSS) is expressed at a substantially higher level compared to splice sites of rank 2 or higher (referred to as minor splice sites, or miSS); within TASS clusters, miSS are expressed several orders of magnitude weaker relative to maSS (Fig 1F, right). We identified the total of 45,739 expressed miSS, the majority of which (82%) were not annotated, unlike the expressed maSS, 87% of which were annotated (Table 2).

To quantify the relative usage of a miSS, we introduced the metric $\varphi$ that takes into account only one end of the split read. It is defined as the number of split reads supporting a miSS as a fraction of the combined number of split reads supporting miSS and maSS. In contrast to the conventional percent-spliced-in (PSI, $\Psi$) metric for exons [30], $\varphi$ measures the expression of a miSS relative to that of the corresponding maSS and takes into account only one end of the supporting split read (S2 Fig).

Since each miSS is associated with a uniquely defined maSS within a TASS cluster, the position of the miSS relative to the position of the maSS, which will be referred to as shift, is defined uniquely. Positive shifts correspond to miSS located downstream of the maSS in a gene, while miSS with negative shifts are located upstream of the maSS. Consistent with previous observations [31], the distribution of shift values for miSS in TASS clusters of size 2 reveals that the most frequent shifts among donor miSS are ±4 nts, while acceptor miSS are most frequently shifted by ±3 nts (Fig 1G, top). These characteristic shifts likely arise from splice site consensus sequences, e.g. NAGNAG acceptor and GYNNGY donor splice sites [17, 32]. Donor miSS in TASS clusters of size larger than 2 are often separated by an even number of nucleotides, while acceptor miSS are often separated by a multiple of 3 nts. For example, rank 2 and rank 3 donor miSS are often located 2 or 4 nts from the maSS and tend to have shifts with opposite signs, while rank 2 and rank 3 acceptor miSS are often separated by 3 nts and tend to be located downstream of the maSS (S3 Fig). In what follows, we refer to a miSS that is located outside of the exon as intronic, and exonic otherwise (Fig 1G, bottom). Intronic miSS correspond to insertions, while exonic miSS correspond to deletions.

In the coding regions, we expect the distance between miSS and maSS to be a multiple of 3 in order to preserve the reading frame. Indeed, shifts by a multiple of 3 nts are the most frequent among coding acceptor miSS, however a considerable proportion of shifts by not a

multiple of 3 nts also occur in both donor and acceptor miSS (S4(A) Fig, top). We therefore asked whether these frame-disrupting shifts are actually expressed, and found that, in spite of their high frequency, the relative expression of miSS in the coding regions, as measured by the $\varphi$ metric, is still dominated by shifts that are multiple of 3 nts (Fig 1G, middle), while the relative expression of miSS in the non-coding regions doesn't depend on the shift (Fig 1G, bottom). Consistent with this, frame-disrupting miSS in the coding regions are significantly upregulated after the inactivation of the nonsense mediated decay (NMD) pathway by co-depletion of two its major components, *UPF1* and *XRN1* [33], while no such upregulation is observed in non-coding regions (Fig 1H). This indicates that the broad positional repertoire of frame-disrupting shifts in coding TASS is efficiently suppressed by NMD.

We next asked whether the expression patterns systematically differ for miSS located upstream and downstream of the maSS. In the coding regions, the acceptor miSS are more often shifted downstream, while miSS located upstream tend to be expressed stronger than miSS located downstream (Fig 1G). This observation supports the splice junction wobbling mechanism, in which the upstream acceptor splice site is usually expressed stronger than the downstream one [18]. However, the expression difference can also be explained by subtle, yet systematic differences in splice site strengths as we found that miSS are on average weaker than maSS (S5(A) Fig), the strength of a miSS relative to the strength of maSS is correlated with its relative expression (S5(B) Fig), and the upstream acceptor miSS tend to be on average stronger than the downstream acceptor miSS (S5(C) Fig, right). Nonetheless, the upstream miSS are expressed stronger than the downstream miSS even for miSS that are similar in strength to their corresponding maSS (S5(D) Fig).

## Expression of miSS in human tissues

Tissue specificity is commonly considered as a proxy for splicing events to be under regulation [5, 34]. To assess the tissue-specific expression of miSS, we calculated the $\varphi_t$ metric, i.e. the relative expression of miSS with respect to maSS, by aggregating GTEx samples within each tissue *t*. However, different tissues are represented by a different number of individuals and, consequently, TASS in different tissues have different read support. To account for this difference and for the dependence of the relative expression of miSS on the gene expression level [17, 35], we constructed a zero-inflated Poisson linear model that describes the dependence of miSS-specific read counts ($r_{miSS}$) on maSS-specific read counts ($r_{maSS}$). Using this model, we estimated the statistical significance of miSS expression and selected significantly expressed miSS using Q-values with 5% threshold [36] and additionally required $\varphi_t$ value to be above 0.05 (Fig 2A). In what follows, we shortly refer to these significantly expressed miSS as significant.

Out of 45,739 expressed miSS, 9,303 (20%) were significantly expressed in at least one tissue (Fig 2B). To identify tissue-specific miSS among significant miSS, we built a linear model with dummy variables corresponding to each tissue (see Methods). A miSS was called tissue-specific if the slope of the dummy variable corresponding to at least one tissue was statistically discernible from zero (Q-value<5%), i.e. the proportion of reads supporting a tissue-specific miSS deviates significantly from the average across tissues. To account not only for significant, but also for substantial changes, we additionally required the absolute value of $\Delta\varphi_t$ be above 0.05, which resulted in a conservative list of 2,496 tissue-specific miSS (Fig 2B and S6(A) Fig). Among these miSS, 234 (9%) became maSS in at least one tissue. In the coding regions, tissue-specific miSS preserve the reading frame more often than non-tissue-specific miSS do (S2 Table); they also have on average stronger consensus sequences and, among the latter, frame-preserving miSS have stronger evidence of translation according to Ribo-Seq data (S6(B) Fig). The intronic region nearby tissue-specific and significantly expressed miSS tends to be more

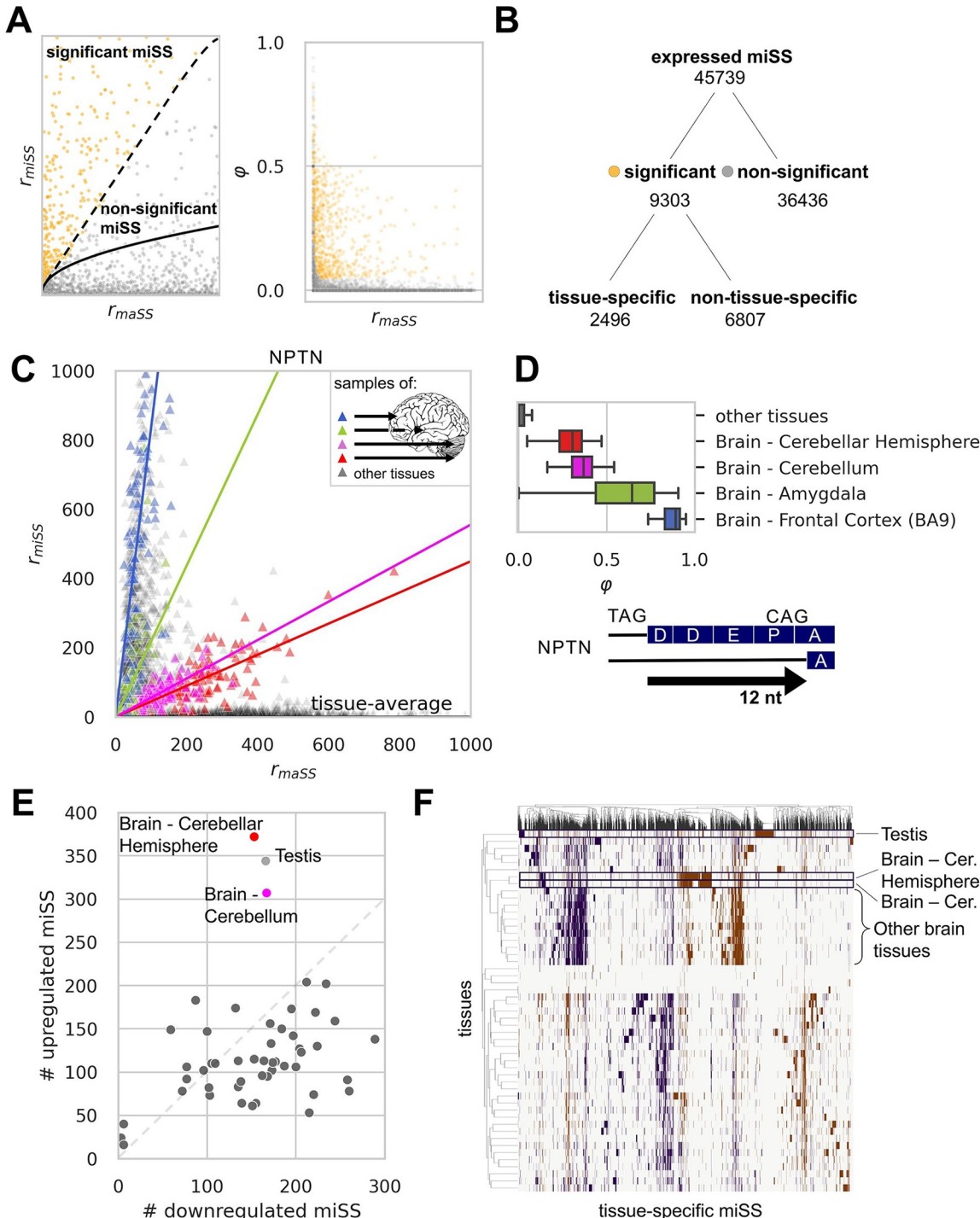

**Fig 2. Expression of miSS in human tissues. (A)** Zero-inflated Poisson model of miSS expression relative to maSS enables identification of significantly expressed miSS (left). Each dot represents a miSS. The expected values of $r_{miSS}$ are shown by the solid curve. FDR cutoff of 5% is shown by the dashed curve. The $\varphi$ value decreases with increase of $r_{maSS}$ (right). **(B)** The classification of expressed miSS. **(C)** Tissue-specific expression of a miSS in the *NPTN* gene. Each dot represents a sample, i.e. one tissue in one individual. Tissue specificity is estimated by a linear model with dummy variables corresponding to tissues. **(D)** The distribution of *NPTN* miSS $\varphi$ values in selected tissues (top). The indel caused by *NPTN* miSS results in the deletion of DDEP motif from the aminoacid sequence (bottom). **(E)** The number of tissue-specific (up- or downregulated) miSS in each tissue. **(F)** The clustering of tissue-specific miSS and tissues based on $\varphi$ values. Blue and red points correspond to downregulated and upregulated miSS, respectively.

conserved evolutionarily as compared to not significant miSS (S6(C) Fig), with frame-disrupting miSS being significantly less conserved than frame-preserving miSS (S6(D) Fig). The distribution of shifts in tissue-specific and other significantly expressed miSS strongly differs from that in non-significantly expressed miSS: among donor miSS, the fraction of +4 shifts is almost two times lower in significantly expressed miSS than in non-significant ones, among acceptor miSS ±3 nt shifts are dominating in significantly expressed miSS while the fraction of other shift variants is lower than in non-significantly expressed miSS (S6(E) Fig).

One notable example of a tissue-specific miSS is in the exon 7 of *NPTN* gene, which encodes neuroplastin, an obligatory subunit of $Ca^{2+}$-ATPase, required for neurite outgrowth, the formation of synapses, and synaptic plasticity [37, 38]. The slope of the linear model has a distinct pattern of variation across tissues, and moreover within brain subregions (Fig 2C). Brain-specific expression of the acceptor miSS instead of maSS in *NPTN* leads to the deletion of Asp-Asp-Glu-Pro (DDEP) sequence from the canonical protein isoform (Fig 2D). Two more examples of tissue-specific and non-tissue-specific miSS are provided in S7 Fig.

Tissues differ by the number of tissue-specific miSS and by the proportion of miSS that are upregulated or downregulated. The sign of the slope in the linear model that describes the dependence of $r_{miSS}$ on $r_{maSS}$ allows to distinguish up- and downregulation. In agreement with previous reports on alternative splicing [39], a number of tissues including testis, cerebellum, and cerebellar hemisphere harbor the largest number of tissue-specific miSS (Fig 2E and S3 Table). The testis and the brain have a distinguished large set of miSS with almost exclusive expression in these tissues that set them apart statistically from the other tissues (Fig 2F).

A special class of TASS are the so-called NAGNAG acceptor splice sites, i.e., alternative acceptor sites that are located 3 bp apart from each other [32]. According to the current reports, they are found in 30% of human genes and appear to be functional in at least 5% of cases [14]. Here, we identified an extended set of 4,761 expressed acceptor miSS, of which 690 are tissue specific, that are located ±3 nt from maSS (S8(A) Fig) which reconfirms 89% of 1,884 alternatively spliced and 29% of 1,338 tissue-specific NAGNAGs reported by Bradley et al [5]. Furthermore, we identified 190 tissue-specific NAGNAGs that are not present in the previous lists [5]. Among them there is a NAGNAG acceptor splice site in the exon 20 of the *MYRF* gene, which encodes a transcription factor that is required for central nervous system myelination. The upstream NAG is upregulated in stomach, uterus and adipose tissues and downregulated in brain tissues (S8(B) Fig). Similarly, we identified an extended set of 2,794 expressed GYNNGY donor splice sites, i.e., alternative donor splice sites that are located 4 bp apart from each other (S8(C) Fig). This set reconfirms 52% of 796 GYNNGY donor splice sites reported by Wang et al [17]. Additionally, we identified 37 novel tissue-specific GYNNGYs including a donor splice site in the exon 2 of the *PAXX* gene (S8(D) Fig), the product of which plays an essential role in the nonhomologous end joining pathway of DNA double-strand break repair [40]. Unlike NAGNAGs, alternative splicing at GYNNGYs disrupts the reading frame and is expected to generate NMD-reactive isoforms [17]. Indeed, GYNNGY miSS along with other frame-disrupting miSS are significantly upregulated after inactivation of the nonsense mediated decay (NMD) pathway by the co-depletion of two major NMD components, *UPF1* and *XRN1* [33] (S8(E) Fig).

The expression of splicing factors is functionally associated with tissue-specific patterns of alternative splicing [41]. In order to identify the potential regulatory targets of splicing factors among miSS, we analyzed the data on shRNA depletion of 103 RNA-binding proteins (RBP) followed by RNA-seq and compared it with tissue-specific expression of miSS and RBP [42]. Our strategy was to identify tissues with significant up- or downregulation of a miSS responding to the inactivation of a splicing factor with the same signature of tissue-specific expression.

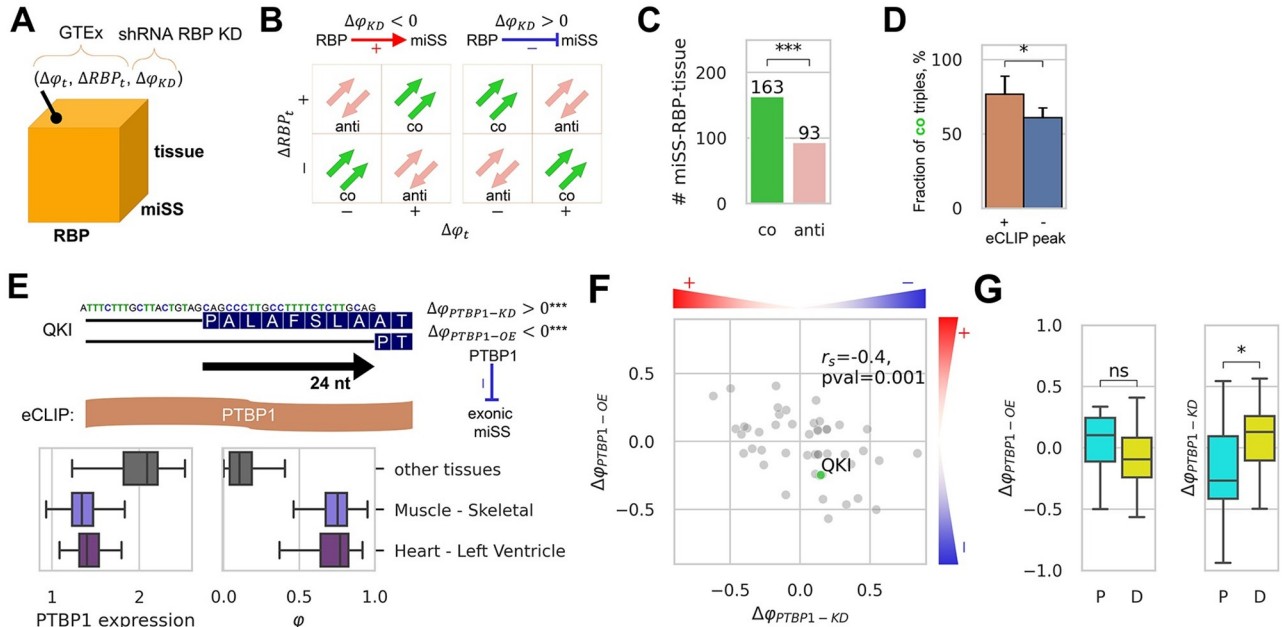

**Fig 3. Regulation of miSS by RBP. (A)** Each miSS-RBP-tissue triple is characterized by three metrics: ($\Delta\varphi_t$, the change of miSS relative usage in tissue $t$; $\Delta RBP_t$, the change of the RBP expression in tissue $t$; and $\Delta\varphi_{KD}$, the change of miSS relative usage upon inactivation of RBP by shRNA-KD. **(B)** The response of miSS to RBP inactivation defines activating ($\Delta\varphi_{KD} < 0$, red) and repressing ($\Delta\varphi_{KD} > 0$, blue) regulation, which together with other metrics define co-directed and anti-directed triples. *(C)* The number of co-directed triples is significantly greater than the number of anti-directed triples. **(D)** The fraction of co-directed miSS-RBP-tissue triples (#*co*/(#*co* + #*anti*)) is significantly greater among triples supported by an eCLIP peak of the RBP near miSS as compared to non-supported triples. **(E)** A deletion of eight aminoacids in the *QKI* gene caused by the exonic miSS overlapping with *PTBP1* eCLIP peak (top). In muscles and heart, the expression of *PTBP1* (log$_2$ *TPM*) is suppressed, while the relative usage of the miSS is promoted (bottom). The miSS is activated in response to *PTBP1* inactivation and is inhibited in response to *PTBP1* overexpression, suggesting downregulation by *PTBP1*. **(F)** The responses of miSS to *PTBP1* overexpression and inactivation by shRNA-KD are negatively correlated. Shown are 50 miSS, in which the response to *PTBP1* overexpression or the response to shRNA-KD of *PTBP1* is statistically significant (Q-value<5%) and substantial (|$\Delta\varphi$|>0.05). **(G)** The responses of proximal (|shift|<5 nt) and distal (|shift|≥5 nt) miSS to *PTBP1* overexpression and inactivation by shRNA-KD are opposite a different mode of regulation for polypyrimidine tracts overlapping the TASS region.

To this end, we identified miSS that are up- or downregulated upon RBP inactivation by shRNA-KD and matched them with the list of tissue-specific miSS and the list of differentially expressed RBP. As a result, we obtained a list of 256 miSS-RBP-tissue triples (see Methods) that were characterized by three parameters, $\Delta\varphi_t$, $\Delta RBP_t$, and $\Delta\varphi_{KD}$, where $\Delta\varphi_t$ is the change of the miSS relative usage in the tissue $t$, $\Delta RBP_t$ is the change of the RBP expression in the tissue $t$, and $\Delta\varphi_{KD}$ is the response of miSS to RBP inactivation by shRNA-KD (Fig 3A). We classified a miSS-RBP-tissue triple as co-directed if the correlation between RBP and miSS expression was concordant with the expected direction of miSS expression changes from shRNA-KD (e.g., if $\Delta\varphi_t > 0$, $\Delta RBP_t > 0$ and $\Delta\varphi_{KD} < 0$) and anti-directed otherwise (Fig 3B). That is, in co-directed triples the direction of regulation from the observed correlation and from shRNA-KD coincide, and in anti-directed triples they are opposite.

In order to obtain a stringent list of regulatory targets, we applied 5% FDR threshold correcting for testing in multiple tissues, multiple RBPs, and multiple miSS, and additionally required that miSS relative usage and RBP expression change not only significantly, but also substantially (|$\Delta\varphi_t$|>0.05, |$\Delta\varphi_{KD}$|>0.05, and |$\Delta RBP_t$|>0.5). As a result, we obtained 163 co-directed and 93 anti-directed miSS-RBP-tissue triples, a proportion that is unlikely to be due to pure chance alone (Fig 3C). Next, we compared our predictions to the footprinting of RBP by the enhanced crosslinking and immunoprecipitation (eCLIP) method [43] and found that co-directed triples are significantly more abundant among miSS-RBP-tissue triples that are

supported by an eCLIP peak (Fig 3D). We summarized the data on co-directed and anti-directed miSS-RBP-tissue triples in (S5 Table). We identified six miSS-RBP candidate pairs with tissue-specific splicing regulation that is co-directed with RBP expression and supported by an eCLIP peak (S6 Table). A notable example is the downregulation of the acceptor miSS in exon 6 of the *QKI* gene by *PTBP1* in muscle and cardiac tissues (Fig 3E), which is consistent with previous reports on the coregulation of alternative splicing by *QKI* and *PTBP1* during muscle cell differentiation [44].

To further investigate potential involvement of *PTBP1* in the regulation of alternative usage of other miSS, we analyzed *PTBP1* overexpression data [45] and identified 50 events, in which the response to *PTBP1* overexpression or the response to shRNA-KD of *PTBP1* was statistically significant (Q-value<5%) and substantial ($|\Delta\varphi|$>0.05) (S7 Table). As expected, the miSS responses to *PTBP1* overexpression and inactivation by shRNA-KD were negatively correlated (Fig 3F). We also found that miSS located proximally (shift < 5) and distally (shift $\geq$ 5) with respect to maSS responded differently to *PTBP1* perturbations. *PTBP1* stimulated the expression of proximal miSS and suppressed the expression of distal miSS (Fig 3G) suggesting a different mode of regulation for polypyrimidine tracts overlapping the TASS region. Such a coordinated expression of TASS has been previously reported in *C. elegans*, in which the expression of proximal and distal TASS is coordinated between germ-line and somatic cells [46], and in human and murine samples, in which the NAGNAG isoforms showed a remarkable co-regulatory pattern [16]. It suggests that PTBP1 could be one of the master regulators that govern such coordinated changes across cell types.

## Expression of miSS in cell types

Tissue-specific alternative splicing originates from that of the constituent cell types, in which TASS splicing programs can also be functionally distinct. To dissect the cell-type-specific expression of TASS, we analyzed RNA-seq data for primary cells from different locations in the human body [47]. Using the same methodology as in the analysis of tissue-specific TASS, we identified 1,821 tissue-of-origin-specific and 1,072 cell-type-specific miSS among significantly expressed miSS (Fig 4A).

Using Pearson correlation as a pairwise similarity measure, we found that miSS expression profiles were more similar for the same cell type from different tissues than for different cell types from the same tissue (Fig 4B). Furthermore, the proportion of co-directed instances among miSS-RBP-cell-type triples is significantly larger than it is among miSS-RBP-tissue triples (Fig 4E). On the one hand, it suggests that miSS expression is governed more by the cell type than by the tissue. It is the case, for instance, for exon 2 of the *IGFLR1* gene, which is upregulated in mesenchymal smooth muscle cells regardless of the tissue (Fig 4C). On the other hand, the expression of some miSS depends on the tissue of origin regardless of the cell type, e.g., a miSS in exon 6 of the *RBM42* gene is upregulated in both heart fibroblasts and heart cardiomyocytes, but not in fibroblasts from other tissues.

## Structural annotation of miSS

Alternative splicing of non-frameshifting TASS results in mRNA isoforms that translate into proteins with only a few amino acids difference. It was reported earlier that alternative splicing tends to affect intrinsically disordered protein regions [48], and that TASS with significant support from ESTs and mRNAs (506 such splice sites in total) are further overrepresented within regions lacking a defined structure [13].

We analyzed the structural annotation of the human proteome (see Methods) and found that significantly expressed miSS preferentially affect disordered protein regions, and tissue-

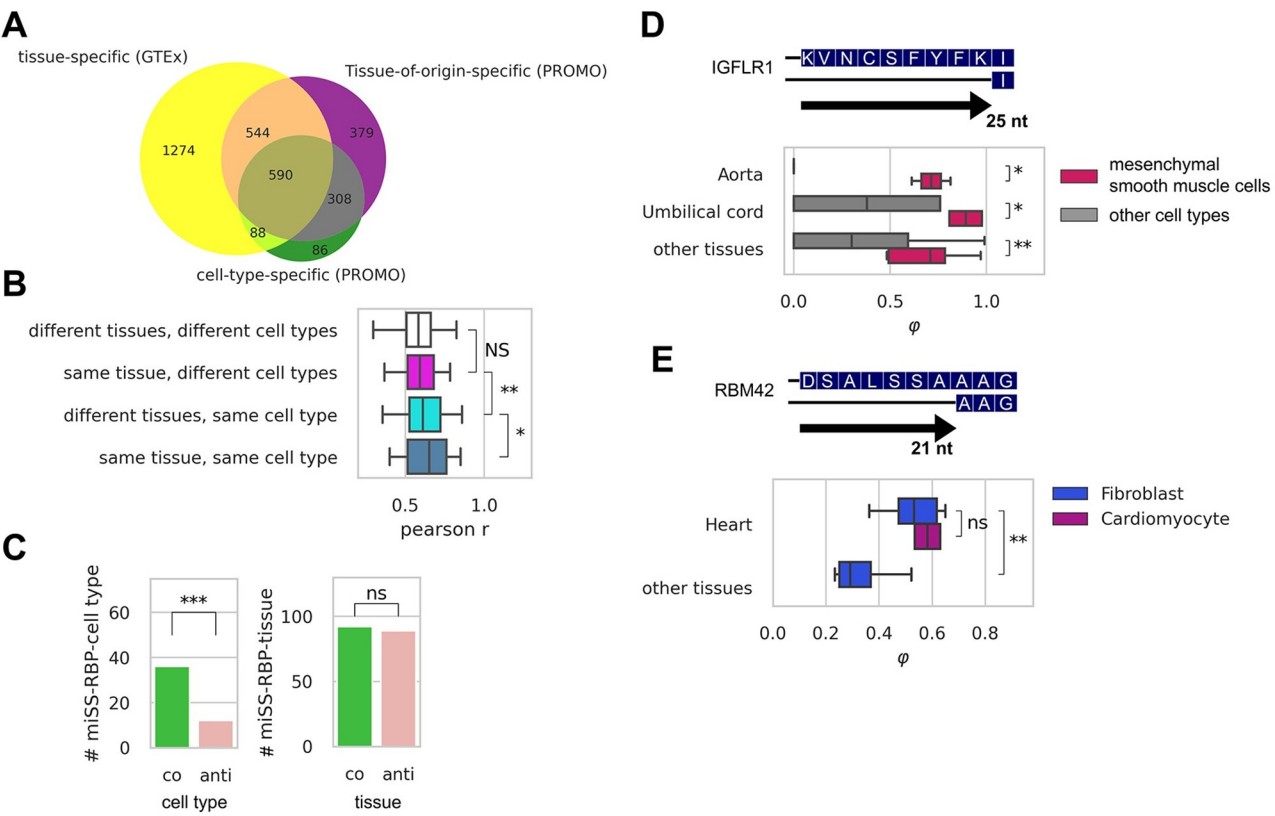

**Fig 4. Expression of miSS in cell types. (A)** The intersection of tissue-specific miSS identified using GTEx data with cell-type-specific miSS and tissue-of-origin-specific miSS identified using PROMO cells data. **(B)** The similarity of miSS expression profiles measured by the Pearson correlation coefficient *r* in the same or different tissues of origin vs. the same or different cell type. **(C).** The abundance of co-directed triples significantly exceeds the abundance of anti-directed triples for the association of miSS-RBP-cell type, while there is no significant difference for the association of miSS-RBP-tissue. **(D)** The expression of an acceptor miSS in exon 2 of the *IGFLR1* gene is upregulated in mesenchymal smooth muscle cells regardless of the tissue-of-origin. **(E)** The expression of an acceptor miSS in exon 6 of the *RBM42* gene is upregulated in both heart fibroblasts and heart cardiomyocytes, but not in fibroblasts from other tissues.

specific miSS are found in disordered regions even more frequently, while the structural annotation of non-significant miSS mirrors that of constitutive splice sites (Fig 5A and S10(B) Fig). Within disordered protein regions, indels that are caused by significantly expressed miSS and their nearby exonic regions are enriched with short linear motifs (SLiMs), short sequence segments that often mediate protein interactions playing important functional roles in physiological processes and disease states [49–51] (Fig 5B). Furthermore, protein sequences of indels and nearby exonic regions for tissue-specific miSS are significantly enriched with methylation sites, one of the most frequent post-translational modifications from the dbPTM database [52]. Interestingly, we found that nucleotide sequences of indels caused by tissue-specific miSS in disordered protein regions are more conserved evolutionarily compared to those of non-tissue-specific miSS, further supporting the enrichment of functional regulatory sites such as SLiMs or PTM (Fig 5D).

A notable example of a SLiM within indel that is caused by miSS is located in the *PICALM* gene, the product of which modulates autophagy through binding to ubiquitin-like *LC3* protein [53, 54] (Fig 5E). The expression of the short isoform lacking 15 nts at the acceptor splice site results in the deletion of Phe-Asp-Glu-Leu (FDEL) sequence, which represents a canonical

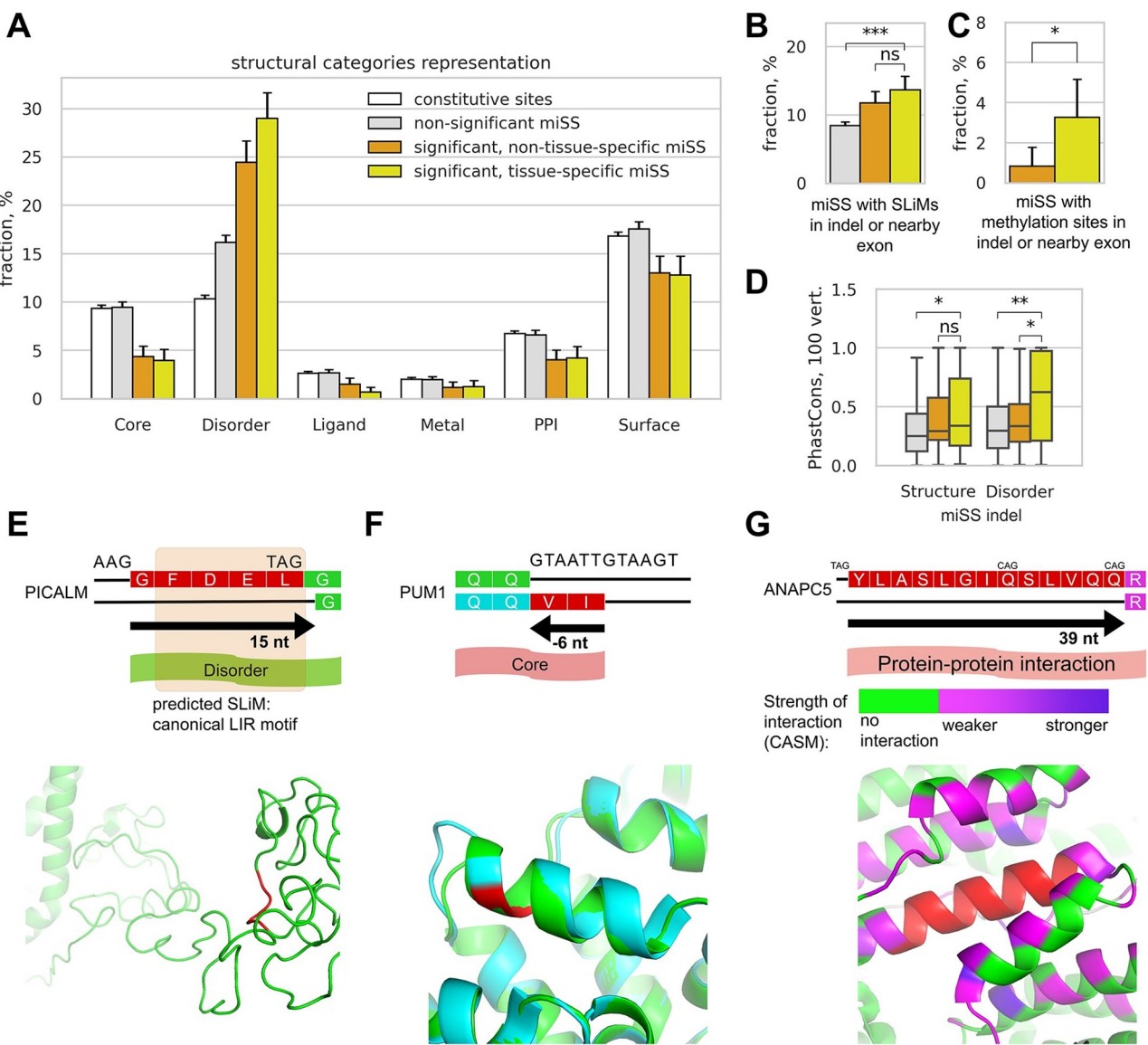

**Fig 5. Structural annotation of miSS. (A)** The proportion of miSS in genomic regions corresponding to protein structural categories. **(B)** The proportion of miSS overlapping occurrences of predicted short linear motifs (SLiMs) from the ELM database. **(C)** The proportion of frame-preserving miSS in genomic regions corresponding to protein methylation sites from the dbPTM database. **(D)** The distribution of the average PhastCons conservation score (100 vertebrates) in the genomic regions between miSS and maSS. **(E)** The expression of an acceptor miSS in the predicted disordered region in the *PICALM* gene results in the deletion of five amino acids containing a predicted canonical LIR motif. The maSS-expressing structure of PICALM was modelled with I-TASSER (green); miSS indel is shown in red. **(F)** The expression of a donor miSS in the *PUM1* gene results in the deletion of two amino acids from the core. The miSS-expressing structure is accessible at PDB (green, PDB ID: 1m8x); the maSS-expressing structure was modelled (cyan) and aligned to the miSS-expressing structure with I-TASSER. **(G)** The expression of an acceptor miSS in the *ANAPC5* gene results in the deletion of 13 amino acids involved in protein-protein interactions. An intermediate splice site (middle CAG) is not significantly expressed. The maSS-expressing structure along with the interacting proteins is accessible at PDB (green, PDB ID: 6TM5); the miSS indel is shown in red. Computational alanine scanning mutagenesis (CASM) in BAlaS [59] identified residues of the neighboring proteins that contribute to the free energy of the interaction with the miSS indel region. The strength of the interaction (the positive change of the energy of interaction) is shown by the gradient color.

LIR (*LC3*-interacting region) motif [55]. This motif interacts with *LC3* protein family members to mediate processes involved in selective autophagy. This miSS is slightly upregulated in whole blood and downregulated in brain tissues consistently with a possible role in physiological regulation of autophagy.

Despite the enrichment of tissue-specific miSS in disordered regions, a sizable fraction of them (more than 10%) still correspond to functional structural categories such as protein core, sites of protein-protein interactions (PPI), ligand-binding, or metal-binding pockets (Fig 5A). We therefore looked further into particular cases to discover novel functional miSS. For example, a shift of the donor splice site by 6 nts in the exon 17 of *PUM1* gene results in a deletion of two amino acids (Fig 5F). This miSS is upregulated in skin, thyroid, adrenal glands, vagina, uterus, ovary, and testis, but downregulated in almost all brain tissues. Only the structure of the miSS-expressing isoform of *PUM1* is accessible in PDB (PDB ID: 1m8x) [56], in which the deletion site maps to an alpha helix. We modelled the structure of maSS-expressing isoform using the I-TASSER web server [57] and found that the alpha helix is preserved, and only its raster is shifted by two amino acids into the preceding loop. The residues in this part of the helix become more hydrophobic, which may influence the overall helix or protein stability. To confirm this, we estimated the stability of the proteins corresponding to the two isoforms using FoldX [58]. The difference of + 13 kcal/mol between the estimated free energies of the minor and the major isoforms indicates that the minor isoform is less stable due to deletion of two hydrophobic residues.

The expression of the miSS in exon 10 of *ANAPC5* gene results in a 13 amino acids deletion from a protein interaction region (Fig 5G). We modelled the interaction of these 13 amino acids with the adjacent protein structures using the computational alanine-scanning mutagenesis (CASM) in BAlaS [59]. We found 58 residues (49 residues in the *ANAPC5* protein and 9 residues in the *ANAPC15* protein), which, when mutated to alanine, cause a positive change in the energy of interaction with the 13 amino acid miSS indel region. The miSS is expressed concurrently with the maSS except for the brain tissues, in which the miSS is significantly downregulated. This may indicate the role of the miSS in various pathways in which *ANAPC5* is involved as an important component of the cyclosome [60, 61].

In order to visualize structural classes associated with TASS, we created a track hub supplement for the Genome Browser [62] (see Supplementary information). The hub consists of three tracks: location of TASS indels, structural annotation of a nearby region, and tissue specific expression of selected TASS (S14 Fig). The catalogue of expressed miSS is also available in the table format (S8 Table).

## Evolutionary selection and conservation of miSS

In order to measure the strength of evolutionary selection acting on significantly expressed and tissue-specific miSS, and to evaluate how it compares with the evolutionary selection acting on maSS and splice sites outside TASS clusters, we applied a previously developed test for selection on splice sites [63] that utilizes confidence limits for the ratio of two binomial proportions based on likelihood scores [64]. We reconstructed the genome of a human ancestor taking marmoset and galago genomes as a sister group and an outgroup, respectively (see Methods). Using canonical consensus sequences of constitutive splice sites, we classified each nucleotide variant at each position as either consensus (Cn) or non-consensus (Nc) nucleotide (S11(A) and S11(B) Fig). Then, we compared the frequency of Cn-to-Nc (or Nc-to-Cn) substitutions at different positions relative to the splice site (observed) with the background frequencies of the corresponding substitutions in neutrally-evolving intronic regions (expected) (S11(C) Fig). The ratio of observed to expected (*O/E*) equal to one indicates neutral evolution (no selection); $O/E > 1$ indicates positive selection; $O/E < 1$ indicates negative selection.

In the coding regions, the strength of negative selection acting to preserve Cn nucleotides in significantly expressed and tissue-specific miSS is comparable to that in maSS and in constitutive splice sites, while no statistically discernible negative selection was detected in miSS that

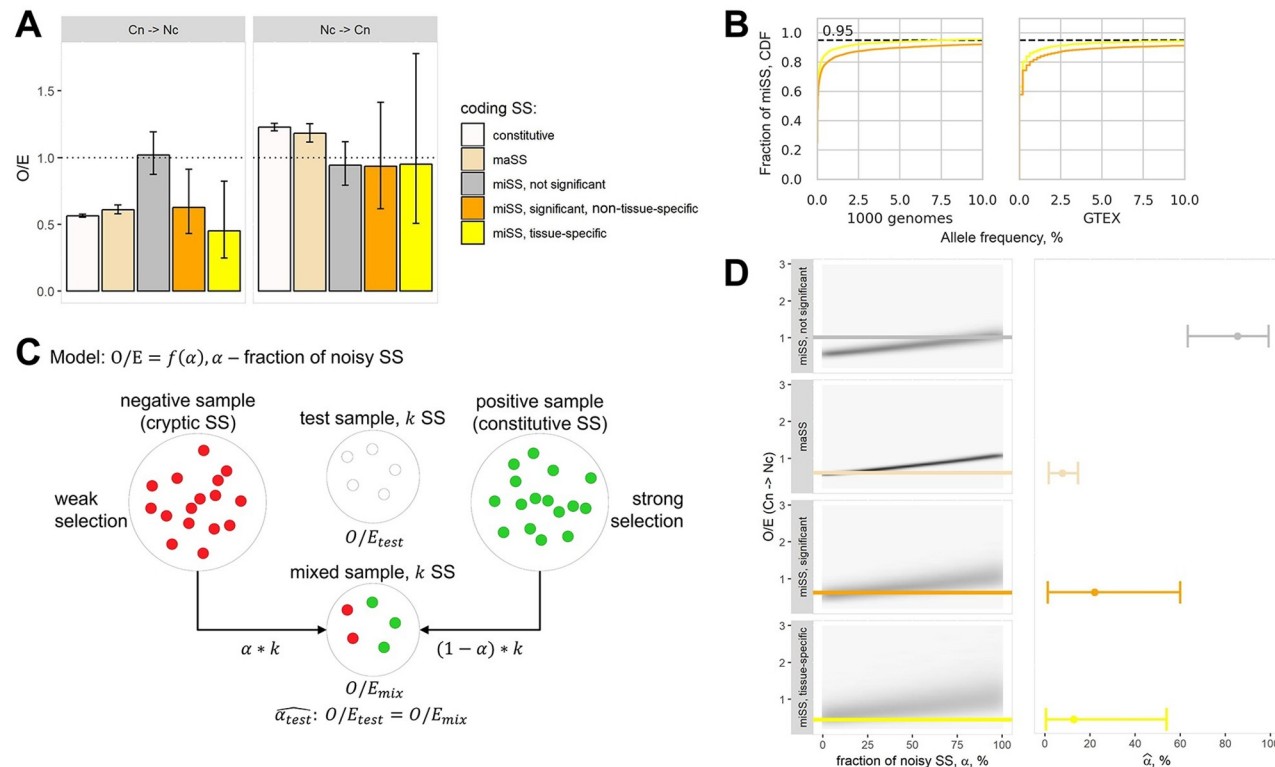

**Fig 6. Evolutionary selection of miSS. (A)** The strength of the selection, defined as the ratio of the observed ($O$) to the expected ($E$) number of substitutions, in selected categories of splice sites in the coding regions. The neutral expectation ($O/E = 1$) is marked by a dashed line. The error bars denote confidence intervals for the ratio of two binomial proportions based on likelihood scores [64]. **(B)** The cumulative distribution of allele frequencies of SNPs in splice site consensus sequences and in 35-nt exonic regions adjacent to splice sites from GTEx and 1000 Genomes projects. **(C)** The mixture model for the estimation of the fraction of noisy splice sites ($\alpha$) using $O/E$ ratio. A test sample of size $k$ is modelled as a mixture of $\alpha k$ purely noisy (cryptic) splice sites and $(1 - \alpha)k$ purely functional constitutive splice sites. **(D)** The estimation of the fraction of noisy splice sites ($\alpha$) from the observed values of $O/E$. Bootstrapped joint distribution of $O/E$ and $\alpha$ values (left). 2.5%, 50% and 97.5% quantiles of the estimated $\alpha$ (right).

are not significantly expressed (Fig 6A, left). In contrast, the strength of positive selection, i.e., the $O/E$ ratio for substitutions that create Cn nucleotides, is not significantly different from 1 in all miSS regardless of their expression, while a significant positive selection was detected in maSS and constitutive splice sites (Fig 6A, right). This indicates that the evolutionary selection may preserve the suboptimal state of significantly expressed and tissue-specific miSS relative to its corresponding maSS. Interestingly, we found that tissue-specific miSS have slightly lower allele frequency of single nucleotide polymorphisms in their splice site sequences and nearby exonic regions compared to non-tissue-specific miSS (Fig 6B) indicating stronger selection acting on the nucleotide sequences of tissue-specific miSS in short-term evolutionary processes [10, 65].

It was shown previously that the strength of the consensus sequence impacts evolutionary selection acting on a splice site [66, 67]. Indeed, the comparison of ancestral consensus sequences showed that constitutive splice sites and maSS have similar ancestral strengths, while miSS are considerably weaker (S11(F) Fig). To control for the influence of the splice site strength on the evolutionary selection, we sampled constitutive splice sites matching them by the ancestral strength with maSS and with miSS (S11(G) Fig). However, despite a considerable difference in strengths, we observed no significant difference in evolutionary selection between constitutive splice sites that were matched to maSS and to miSS, indicating that the observed

difference in selection acting on miSS and maSS is not due to weaker consensus sequences of miSS.

The difference in evolutionary selection between significantly expressed and the rest of miSS could arise from the difference in the fraction of noisy splice sites in these miSS categories. To estimate the fraction of noisy splice sites ($\alpha$), we constructed a mixture model (Fig 6C), in which we combined $\alpha k$ splice sites from the negative set of cryptic splice sites and $(1-\alpha)k$ splice sites from the positive constitutive set and measured the strength of evolutionary selection in the combined sample for all values of $\alpha$. Using this model, we constructed the joint distributions of $\alpha$ and $O/E$ values of Cn-to-Nc substitutions for maSS, significantly expressed non-tissue-specific miSS, tissue-specific miSS and the rest of miSS (Fig 6D, left). From these distributions, we estimated 95% confidence intervals for the values of $\alpha$ that correspond to the actual $O/E$ values in the observed samples (Fig 6D right and S11(H) Fig). The resulting estimates for the fraction of noisy splice sites among maSS, significantly expressed non-tissue-specific miSS, tissue-specific miSS, and the rest of the miSS are <15%, <60%, <54% and >63%, respectively, indicating that at least 46% of tissue-specific miSS are statistically discernible from noise.

## Discussion

Increasing amounts of high-throughput RNA-seq data have uncovered the expanding landscape of human alternative splicing [68]. Here, we present the most complete up-to-date catalogue of 45,739 miSS, of which 9,303 are significantly expressed in healthy human tissues according to GTEx data. It significantly extends the TASSDB2 database constructed based on the evidence from ESTs [24] by adding ~18k miSS (S12(A) Fig), which are enriched with significantly expressed miSS despite being weakly expressed on average (S12(B) and S12(C) Fig). It also adds data on specific TASS classes such as NAGNAGs [5] and GYNNGYs [17]. On the one hand, the number of detected TASS is reaching a plateau with increasing the number of GTEx samples up to 8,548 (S13 Fig) indicating that this catalogue is close to being complete. On the other hand, a substantial fraction of TASS are noisy (more than 63% among miSS that are not significantly expressed) reflecting natural tradeoff between sensitivity and specificity.

While the majority of miSS in the coding regions are located downstream of their respective maSS, the upstream miSS tend to be expressed stronger, i.e., the spliceosome tends to systematically choose a miSS that is located upstream. This pattern likely results from the linear scanning mechanism, in which the spliceosome traverses the pre-mRNA in the 5' to 3' direction so that tandem splice sites follow first come-first served principle [69, 70]. The relative expression is the strongest for the frame-preserving acceptor miSS in support of the observation that transcript isoforms with frame-disrupting miSS are suppressed by NMD. We therefore expected that frame-disrupting miSS would be rare among significantly expressed and tissue-specific miSS. However we observe that almost a half of tissue-specific coding miSS disrupt the reading frame (S2 Table). Furthermore, frame-disrupting tissue-specific miSS are more conserved than non-significantly expressed miSS (S6(D) Fig) indicating a potential function such as, for example, fine-tuning of gene expression levels via NMD [71–73].

Previous reports indicated that strongly expressed miSS located at a distance of 3, 6, 9 nt from the maSS in coding regions, which are to a large extent equivalent to the significantly expressed miSS introduced here, are overrepresented in disordered protein regions [13]. The evolutionary selection against alternatively spliced NAGNAGs in protein-coding genes is stronger in structured regions than in disordered regions [13]. Here, we extended this result by showing that tissue-specific miSS are even more enriched in disordered protein regions than other significantly expressed miSS with the most pronounced enrichment among

acceptor exonic miSS, which lead to the deletion in the protein sequence. Furthermore, we showed that tissue-specific miSS are associated with SLiMs and post translational modification sites. While there is no positive selection for Cn nucleotides among neither significantly expressed nor other miSS, we observed a strong negative selection acting to preserve Cn nucleotides in the former (Fig 6A). This finding can be explained by the tendency of functional miSS to preserve the suboptimal state relative to maSS, i.e functional miSS are evolutionarily conserved and maintain their Cn nucleotides, but they also do not harbor more Cn nucleotides not to outcompete maSS. Furthermore, we showed a tendency of many tissue-specific miSS to be regulated by RBP, e.g., the miSS in the exon 6 of the *QKI* gene is likely regulated by *PTBP1* (Fig 3E). All these findings are indicative of a functional role of at least a proportion of miSS.

It has been demonstrated that cell-type-specific alternative splicing within the same tissue may affect a large fraction of multi-exon genes and govern cell fate and tissue development [74, 75]. For example, pairs of exons in genes *Gria1* and *Gria2* follow a strict mutually-exclusive pattern between different neuronal types [76]. In this study, we examined miSS expression in primary cells from different tissues and identified hundreds of cell-type-specifically expressed miSS. While the comparison of expression profiles suggested that miSS expression and its regulation by RBPs depends to a greater degree on the cell type than on the tissue of origin, both local (cell type) and global (tissue) gene expression environments can contribute to the specificity of miSS usage (Fig 4D and 4E).

The observations made in this manuscript are based on the analysis of RNA-seq data from the GTEx project [26]. It is hence worthwhile to address the question which proportion of these alternative splicing events translate to the protein level. Direct measurement of this proportion by, for example, shotgun proteomics is not instructive for many reasons, including limited coverage and low sensitivity of such experiments [77], as well as the fact that the cleavage site consensus of a widely used trypsin protease overlaps with the amino acid sequence induced by the splice site consensus, thus producing non-informative peptides [78]. This question has been debated in the literature [21–23]. On the one hand, proteomics data support the expression of a single predominant protein isoform for most human genes [21]. On the other hand, ribosome profiling suggests translation of alternative isoforms [79], and experimental studies demonstrate the functional importance of alternative splicing in modulation of protein-protein interactions [80]. Our study adds to this debate in that we have collected multiple lines of evidence that support expression on the protein level and functional importance of TASS-related isoforms. Our estimate of significantly expressed miSS largely exceeds the conservative estimate of proteomics-supported alternative splicing events [21]. The analysis of Ribo-Seq experiments supports their expression, and in many cases this expression is tissue-specific. We also showed that significantly expressed miSS, as well as maSS, are under negative selection pressure. Finally, our analysis confirms that sites in protein sequence that correspond to TASS events are depleted from structured protein regions, just as for alternative splicing events in general [48, 81], which also suggests their non-neutral evolution and hence existence on the protein level. In line with previous research [81], we demonstrated that when located in disordered protein regions, TASS-associated events often affect sites of post-translational modification.

## Conclusion

Tandem alternative splice sites (TASS) are the second most abundant subtype of alternative splicing. The analysis of a large compendium of human transcriptomes presented here has uncovered a large and heterogeneous dataset of TASS, of which a significant fraction are expressed above the noise level and have signatures of tissue-specificity, evolutionary selection,

conservation, and regulation by RBP. This suggests that the number of functional TASS in the human genome may be larger than it is currently estimated from proteomic studies, while the majority of TASS represent splicing noise.

## Materials and methods

### The catalogue of TASS

**The annotated splice sites.**    Throughout this paper, we use GRCh37 (hg19) assembly of the human genome which was downloaded from the UCSC genome browser [82]. To identify the annotated splice sites, we extracted internal boundaries of non-terminal exons from the comprehensive annotation of the GENCODE database v19 [27] and from UCSC RefSeq database [28]. As a result, we obtained 569,694 annotated splice sites (S1(A) Table).

**Expressed splice sites.**    The RNA-seq data from 8,548 samples in the Genotype-Tissue Expression (GTEx) consortium v7 data was analyzed as before [26]. Short reads were mapped to the human genome using STAR aligner v2.4.2a by the data providers [83]. Split reads supporting splice junctions were extracted using the IPSA package with the default settings [30] (Shannon entropy threshold 1.5 bit). At least three split reads in at least two samples from different tissues were required to call the presence of a splice site. Samples of EBV-transformed lymphocytes and transformed fibroblasts and three samples with aberrantly high number of split reads were excluded. Only split reads with the canonical GT/AG dinucleotides were considered. Germline polymorphisms (SNPs, deletions and insertions) located within the splice site or within 35 nt of adjacent exonic regions were identified. Splice sites that were expressed exclusively in the samples, in which a polymorphism was present but absent in the other samples, were excluded to avoid split read misalignment caused by the discrepancy between the reference genome and the individual genotypes. This filtration removed 1.15% of expressed splice sites that were supported by 0.3% of the total number of split reads. As a result, we obtained 794,646 expressed splice sites (S1(A) Table).

**Cryptic splice sites.**    We used SpliceAI software [29] to scan the canonical transcriptome sequences assembled by authors and selected splice sites with splice probability score greater than 0.1. According to the data provided by the authors, at least 95% of exons having $\Psi$ value below 0.1 are flanked by splice sites that fall below this score threshold. Splice sites that were previously called expressed or annotated were excluded resulting in a list of 607,639 cryptic splice sites (S1(A) Table).

**Categorization of splice sites within TASS clusters.**    A TASS cluster (and all splice sites within it) was categorized as coding if it contained at least one non-terminal boundary of a coding exon, and non-coding otherwise. Thus, non-coding splice sites are located in untranslated regions (UTRs) of protein-coding genes or in other gene types such as long non-coding RNA. Splice sites were ranked based on the total number of supporting split reads. The splice site strength was assessed by MaxEntScan software [84] which computes a similarity of the splice site sequence and the consensus sequence. The higher MaxEnt scores correspond to splice site sequences that are closer to the consensus.

### Response of TASS clusters to NMD inactivation

To assess the response of TASS clusters to the inactivation of NMD, we used RNA-seq data from the experiments on co-depletion of *UPF1* and *XRN1*, two key components of the NMD pathway [33]. Short reads were mapped to the human genome using STAR aligner v2.4.2a with the default settings. The read support of splice sites was called by IPSA pipeline as before (see processing of GTEx data). TASS in which the major splice site was supported by less than 10 reads were discarded. The response of a miSS to NMD inactivation was measured by $\varphi_{KD} -$

$\varphi_C$, where $\varphi_{KD}$ is the relative expression in KD conditions and $\varphi_C$ is the relative expression in the control.

## Expression of miSS in human tissues

**Significantly expressed (significant) miSS.** The number of reads supporting a splice site can be used for presence/absence calls, however it depends on the local read coverage in the surrounding genomic region and on the total number of reads in the sample [35, 85]. A good proxy for these confounding factors is the number of reads supporting the corresponding maSS. We therefore quantified the expression of miSS relative to maSS and selected miSS that are expressed at significantly high level at the given maSS expression level, separately in each tissue. Since the number of reads often exhibits an excess of zeros, we treated the total number of reads supporting a miSS ($r_{miSS}$) in each tissue as a zero-inflated Poisson random variable with the parameters $(\hat{\pi}(r_{maSS}), \hat{\lambda}(r_{maSS}))$ that depend on the number of reads supporting the corresponding maSS ($r_{maSS}$) as follows:

$$\hat{\lambda} = a_0 r_{maSS}^{a_1} \tag{1}$$

$$\hat{\pi} = \text{logit}^{-1}(b_0 + b_1 r_{maSS}). \tag{2}$$

We estimated the parameters $a_0$, $a_1$, $b_0$, and $b_1$ separately in each tissue using zero-inflated Poisson (ZIP) regression model [86], computed the expected value of $r_{miSS}$ for each miSS given the value of $r_{maSS}$, and assigned a P-value for each miSS as follows:

$$P-value = 1 - (CDF_{Poisson}(r_{min}, \hat{\lambda})(1 - \hat{\pi}) + \hat{\pi}). \tag{3}$$

To account for multiple testing, we converted the matrix of P-values for all miSS in all tissues to a linear array and estimated the false discovery rate by Q-value [36]. A miSS was called significantly expressed (or shortly significant) if it had the Q-value below 5% and $\varphi$ value greater than 0.05 in at least one tissue.

**Tissue-specific miSS.** The level of expression of a miSS relative to its corresponding maSS is reflected by the $\varphi$ metric. To identify tissue-specific miSS among significantly expressed miSS, we analyzed the variability of the $\varphi$ metric between and within tissues using the following linear regression model. For each significant miSS individually, we model $r_{miSS}$ as a function of $r_{maSS}$ by the equation

$$r_{miSS} = a_0 r_{maSS} + \sum_t a_t D_t r_{maSS}, \tag{4}$$

where $D_t$ is a dummy variable corresponding to the tissue $t$. The slope $a_t$ in this model can be interpreted as the change of the miSS relative usage in tissue $t$ with respect to the tissue average as follows

$$\hat{\varphi}_{tissue-average} = \frac{\hat{a}_0}{1 + \hat{a}_0} \tag{5}$$

$$\hat{\varphi}_t = \frac{\hat{a}_0 + \hat{a}_t}{1 + \hat{a}_0 + \hat{a}_t} \tag{6}$$

$$\Delta\hat{\varphi}_t = \frac{\hat{a}_t}{(1 + \hat{a}_0 + \hat{a}_t)(1 + \hat{a}_0)} \tag{7}$$

The significance of tissue-specific changes of $\varphi$ represented by $a_t$ can also be estimated using this linear model. This allows assigning P-values (and Q-values) to $a_t$ for each miSS in each tissue. In order to filter out significant, but not substantial changes of tissue-specific miSS expression, we required the Q-value corresponding to $a_t$ be below 5% and the absolute value of $\Delta\hat{\phi}_t$ be above 5%; a miSS satisfying these conditions was called tissue-specific in the tissue $t$. A miSS was called tissue-specific if it was specific in at least one tissue. Additionally, the sign of $a_t$ allows to distinguish upregulation ($a_t > 0$) or downregulation ($a_t < 0$) of a miSS in the tissue $t$.

## Regulation of miSS by RBP

RNA-seq data from the experiments on the depletion of 248 RBPs in two human cell lines (K562 and HepG2) were downloaded from ENCODE portal website in BAM format [87]. Short reads were mapped to the human genome using STAR aligner v2.4.0k [83]. Out of 248 RBPs, we left only those for which 8 samples were present: two KD and two control samples for each of the two cell lines. Additionally, we required the presence of at least one publicly available eCLIP experiment [43] for each RBP. This confined our scope to 103 RBPs (S4 Table).

We used rMATS-turbo v.4.1.0 [88] in novelSS mode to identify both novel and annotated alternative splicing events between KD and control samples for each RBP in each cell line. The minimum intron length and the maximum exon length were set to 10 and 1000, respectively. Since the definition of $\Psi$ value for alternative donor and acceptor splice sites in rMATS pipeline corresponds to the definition of $\varphi$ value, we used the rMATS output to directly extract $\Delta\varphi_{KD}$ values and P-values. We obtained Q-values for 9,303 significantly expressed miSS in each RBP and cell line. In order to filter out significant, but not substantial changes of miSS expression between KD and control samples, we required the Q-value be below 5% and the absolute value of $\Delta\varphi_{KD}$ be above 0.05 in both HepG2 and K562 cell lines. As a result, we obtained 221 significant RBP-miSS pairs, of which 65 pairs (29%) showed a discordant response to KD between cell lines (S9 Fig). These cases were excluded, and 156 RBP-miSS pairs (101 pairs with $\Delta\varphi_{KD} > 0$ and 55 pairs with $\Delta\varphi_{KD} < 0$) were kept for downstream analysis of miSS-RBP-tissue triples.

The gene read counts data was downloaded from GTEx (v7) portal on 08/05/2020 [89] and processed by DESeq2 package using apeglm shrinkage correction [90]. Differential expression analysis was done for each tissue against all other tissues. The P-values for 103 RBPs in each tissue were adjusted for FDR using Q-value [36]. An RBP was classified as tissue-specific if the Q-value in the corresponding tissue was below 5% and the absolute value of $\log_2$ fold change was higher than 0.5. A tissue-specific RBP was considered upregulated in tissue $t$ ($\Delta RBP_t > 0$) if the $\log_2$ fold change value was positive and downregulated ($\Delta RBP_t < 0$) otherwise. As a result, we obtained 1,115 RBP-tissue pairs (388 upregulated pairs and 727 downregulated pairs).

We obtained 14,005 miSS-tissue pairs (6,265 upregulated pairs and 7,740 downregulated pairs) in the analysis of tissue-specific expression of miSS (see Tissue-specific miSS section). The intersection of these pairs with RBP-tissue pairs and RBP-miSS pairs resulted in 256 miSS-RBP-tissue triples. Each triple was classified as co-directed or anti-directed according to the rules shown in Fig 3B.

The eCLIP peaks, which were called from the raw data by the producers, were downloaded from ENCODE data repository in bed format [91, 92]. The peaks in two immortalized human cell lines, K562 and HepG2, were filtered by the condition $\log FC \geq 3$ and P- value $< 0.001$ as recommended [43]. Since the agreement between peaks in the two replicates was moderate (the median Jaccard distance 25% and 28% in K562 and HepG2, respectively), we took the

union of peaks between the two replicates in two cell lines, and then pooled the resulting peaks. The presence of eCLIP peaks was assessed in the ±20 nt vicinity of a miSS position.

We downloaded the *PTBP1* overexpression data [45] (2 full-length *PTBP1* overexpression samples, 4 control samples) from NCBI SRA archive in fastq format under the accession number SRP059242. Short reads were mapped to the human genome using STAR aligner v2.4.2a with the default settings. We used rMATS-turbo v.4.1.0 with the same approach as we used for shRNA-KD data to infer $\Delta\varphi_{PTBP1-OE}$ values and associated P-values and Q-values for 9,303 significantly expressed miSS.

## Expression and regulation of miSS in primary cells

Primary cell transcriptome data (94 RNA-seq experiments) from 19 tissues of origin were downloaded from ENCODE portal website in BAM format [47, 87]. Each sample was assigned to one of the 9 cell types (mesenchymal smooth muscle cells, endothelial cells, epithelial cells, cardiomyocytes, fibroblasts, melanocytes, stem cells, preadipocytes, skeletal muscle cells) according to metadata. Short reads were mapped to the human genome by the data providers using STAR aligner v2.3.1z [83]. The read support of splice sites was called by IPSA pipeline as before (see processing of GTEx data). The identification of cell-type-specific and tissue-of-origin-specific miSS was done using linear regression as before (see identification of tissue-specific miSS in GTEx data). The $\varphi$ values were calculated for 9,303 significantly expressed miSS in each sample requiring at least one of $r_{miSS}$ and $r_{maSS}$ values to be greater than 20 for positive $\varphi$ values and substituting the $\varphi$ values with zero otherwise. Pearson correlation coefficient was used as a measure of similarity of miSS expression profiles. Gene expression profiles were assessed by the data providers using RSEM v.1.2.19 [93]. Read counts were library size-corrected using the DESeq2 package [94]. From the gene set, we selected 103 RBPs introduced before (see regulation of miSS by RBP). MiSS-RBP-cell type triples and miSS-RBP-tissue triples were obtained as before by merging PROMO miSS and RBP expression data with the responses of miSS to shRNA-KD of RBP. A miSS-RBP-cell type (miSS-RBP-tissue) triple was defined co-directed if the sign of the Spearman correlation coefficient of miSS expression and RBP expression within samples of this cell type (tissue) coincided with the expected sign of correlation inferred from shRNA-KD of RBP. For example, if the miSS is upregulated in the KD of RBP, the expected sign of the correlation would be negative.

## Evidence of miSS translation in Ribo-Seq data

The global aggregate track of Ribo-Seq profiling, which tabulates the total number of footprint reads that align to the A-site of the elongating ribosome, was downloaded in bigWig format from GWIPS-viz Ribo-Seq genome browser [95]. It was intersected with TASS coordinates to obtain position-wise Ribo-Seq signal for miSS and maSS. The analysis was carried out on intronic miSS in TASS clusters of size 2. For each miSS, the relative Ribo-Seq support was calculated as

$$RS = \frac{\#reads_{miSS}}{\#reads_{miSS} + \#reads_{maSS}}, \tag{8}$$

where $\#reads_{miSS}$ and $\#reads_{maSS}$ are the number of Ribo-Seq reads supporting the first exonic nucleotide of miSS and maSS, respectively. Higher values of *RS* indicate stronger evidence of translation.

## Structural annotation of miSS

All amino acids that are lost or gained due to using miSS instead of maSS were structurally annotated with respect to their spatial location in protein three-dimensional structure using

StructMAn [96]. As a control, we also annotated all amino acids in all isoforms of the human proteome. Briefly, the procedure of structural annotation consists in mapping a particular amino acid into all experimentally resolved three-dimensional structures of proteins homologous to a given human isoform. The mapping is done by means of pairwise alignment of the respective protein sequences. Then the spatial location of the corresponding amino acid residue in the structure is analyzed in terms of proximity to other interaction partners (other proteins, nucleic acids, ligands, metal ions) and propensity to be exposed to the solvent or be buried in the protein core. Such annotations from different homologous proteins are then combined taking into account sequence similarity between the query human isoform and the proteins with the resolved structures, alignment coverage and the quality of the experimental structure. This resulted in structural annotations for 23,095,050 amino acids from 88,573 protein isoforms.

To use the structural annotation of amino acids in the analysis of TASS, we established a correspondence between 86,647 UniProt protein identifiers and 106,403 ENSEMBL transcripts identifiers discarding 3,194 transcripts that had ambiguous mappings [97]. We used custom scripts to map 23,095,050 amino acids within structural annotation of UniProt entries to the human transcriptome and, furthermore, to the human genome using ENSEMBL transcript annotation. This procedure yielded 17,093,614 non-redundant genomic positions since some UniProt entries correspond to alternative isoforms of the same protein, and thus some amino acids from different entries can map to the same nucleotide in the genome. At that, positions that had ambiguous structural annotation from different transcripts were discarded.

Unlike maSS and exonic miSS, most of the intronic miSS are located outside of ENSEMBL transcripts and thus can not be directly classified based on the structural annotation. However, the structural annotation of exonic miSS coincides with that of the respective maSS in most cases (S10(A) Fig). We therefore assumed that the short distance between maSS and miSS allows to assign the structural annotation of the first exonic nucleotide of a maSS to all miSS including miSS located in introns. This way we defined structural annotation for 6,879 out of 12,667 frame-preserving expressed miSS in coding regions.

SLiM protein coordinates were mapped to genomic coordinates as described above. Regions between maSS and miSS (miSS indels) are compared with nearby exonic regions defined as the regions of the same length as miSS indels but located in the adjacent exons on the distance equal to the indel length. A SLiM is recognized to overlap with a particular region (miSS indel of nearby exonic region) if its genomic projection overlaps at least one exonic nucleotide.

## Evolutionary selection of miSS

Splice sites of annotated human transcripts were extracted from the comprehensive annotation of the human transcriptome (GENCODE v19 and NCBI RefSeq) using custom scripts [27, 28]. Internal boundaries of non-terminal exons (excluding splice sites overlapping with TASS clusters) were classified as constitutive splice sites if they were used as splice sites in all annotated transcripts. Position weight matrices were used to build consensus sequences for donor and acceptor constitutive splice sites as described in [98, 99]. Orthologs of the annotated human splice sites were identified in multiple sequence alignment of 46 vertebrate genomes with the human genome (GRCh37), which were downloaded from the UCSC Genome Browser in MAF format [82]. The alignments with marmoset and galago (bushbaby) genomes were extracted from MAF, and the alignment blocks were concatenated. The genomic sequence of splice sites in the common ancestor of human and marmoset with galago as an outgroup was reconstructed by parsimony [100]. Only splice sites with the canonical GT/AG dinucleotides

in all three genomes were considered. The analysis was further confined to TASS clusters of size 2, in which only intronic miSS were considered to avoid the confounding effect of selection acting on the coding sequence in exonic miSS. This procedure resulted in 34,550 TASS (17,275 maSS and 17,275 miSS) in the coding regions.

To estimate the strength of evolutionary selection acting on Cn and Nc nucleotides, we used a previously developed method with several modifications [63]. First, only intronic positions from the positions +3 to +6 for the donor splice sites and positions from -24 to -3 for the acceptor splice sites were considered (the canonical GT/AG dinucleotides were excluded as they were required to be conserved). The substitution counts were summed over all positions in these ranges. Furthermore, splice sites from the human genome were mapped onto the ancestral genome using MAF alignments but the substitutions were analyzed in the marmoset lineage, where the substitutions process goes independently from the human lineage (S11(B) Fig). This approach mitigates the systematic underrepresentation of Cn-to-Nc substitutions and the overrepresentation of Nc-to-Nc substitutions in the human lineage leading to artificial signs of strong positive and negative selection in cryptic and not significant miSS (S11(D) Fig) [63]. Constitutive splice sites were matched to maSS (miSS) by the ancestral strength using random sampling from the set of constitutive splice sites without replacement and requiring the strength difference not larger than 0.01.

### Allele frequencies of SNPs

Germline SNPs located within [-35 nt, +6nt] for the donor miSS and within [-21 nt, +35 nt] for the acceptor miSS were identified in GTEx and 1000 Genomes [10] data. Allele frequencies of the SNPs were obtained using `vcftools` [101] and custom scripts. For comparison of allele frequencies between different miSS categories, the maximum allele frequency of SNPs related to each of the miSS was calculated.

### Mixture model for the estimation of the fraction of noisy miSS

The mixture model to estimate the fraction of noisy splice sites (Fig 4C) was constructed as follows. Denote by $k$ the size of the sample of interest (tissue-specific miSS, significant non-tissue-specific miSS, non-significantly expressed miSS, or maSS). We assume that the sample of interest is a mixture of two subsamples, $\alpha k$ splice sites from the negative set (cryptic splice sites, which demonstrate no evidence of selection, S11(D) Fig) and $(1 - \alpha)k$ splice sites from the positive set (all constitutive splice sites). For every $\alpha$ in the range from 0 to 1 with the step 0.0033, we sample randomly $\alpha k$ elements from the negative set and $(1 - \alpha)k$ elements from the positive set 300 times and construct the joint frequency distribution of $\alpha$ and $O/E$. To obtain the marginal (conditional) distribution corresponding to the observed value of $O/E$ in the actual set of interest, we use an infinitesimal margin $\epsilon$ to compute the empirical probability density in $(O/E - \epsilon, O/E + \epsilon)$, and take the limit $\epsilon \rightarrow 0$ using the linear regression model $p = \beta_0 + \beta_1 \epsilon$. The quantiles were calculated for every $\epsilon$ in the range from 0.025 to 0.5 with the step 0.005 (S11(H) Fig). The interval estimates of $\alpha$ are inferred from the 2.5% and 97.5% quantiles. The Nc-to-Cn substitutions could also be used to construct a similar model, but their applicability is rather limited as the $O/E$ values of Nc-to-Cn substitutions for all miSS categories are not statistically discernible from one (Fig 6A, right).

### Statistical analysis

The data were analyzed and visualized using R statistics software version 3.4.1 and ggplot2 package, python version 3.7.6 and seaborn package. Non-parametric tests were performed with the statsmodels package using normal approximation with continuity correction. MW

denotes Mann-Whitney sum of ranks test. Error bars in all figures and the numbers after the ± sign represent 95% confidence intervals. One-tailed P-values are reported throughout the paper, with the exception of linear regression models, in which we use two-tailed tests. Levels of significance 0.05,0.01,0.001 are denoted as *,**,***.

## Supporting information

**S1 Fig. The number of annotated, *de novo*, and cryptic TASS in the coding and non-coding regions.** In addition to the numbers provided in the figure, there are 163 cryptic splice sites in lincRNA genes and 580 annotated but not expressed splice sites in lincRNA genes.
(TIF)

**S2 Fig. The definition of $\varphi$ value exemplified.** A hypothetical maSS is supported by 4 split reads, while a hypothetical miSS is supported by 3 split reads, resulting in the $\varphi$ value of 3/7.
(TIF)

**S3 Fig. TASS clusters of size three.** A TASS cluster of size three is characterized by two shift values: the rank two miSS relative to maSS, and rank three miSS relative to maSS. The top panel shows the joint distribution of rank two miSS shift (x-axis) and rank three miSS shift (y-axis) for donors (left) and (acceptors). The bottom panel shows LOGO charts of miSS sequences corresponding to shifts of +4 and -2 for the donor splice site, and +3 and +6 shifts for the acceptor splice site.
(TIF)

**S4 Fig. The distribution of shifts across TASS categories. (A)** Shift frequencies in coding vs. non-coding regions (see Fig 1G for comparison). **(B)** The abundance and relative expression of upstream vs. downstream shifts.
(TIF)

**S5 Fig. The strength of the consensus sequence of TASS. (A)** According to MaxEnt scores, maSS (i.e., rank one sites) are on average stronger than miSS (i.e., rank 2,3,4,5). **(B)** Within each rank group, the relative usage of a miSS ($\varphi$) generally increases with increasing $\Delta$ MaxEnt value, its strength relative to that of the maSS. **(C)** The distribution of $\Delta$ MaxEnt values for upstream and downstream shifts. **(D)** The relative usage of a miSS ($\varphi$) as a function of the absolute difference of TASS strengths. The upstream miSS are used more frequently when the splice sites are nearly of the same strength.
(TIF)

**S6 Fig. Features of significantly expressed and tissue-specific miSS. (A)** Tissue-specific miSS are defined to have $|\Delta\varphi_t|>0.05$ (x-axis) and Q-value $<0.05$ (y-axis) in at least one tissue. **(B)** The distribution of $\Delta$ MaxEnt values for miSS in different expression categories (left). The distribution of *RS* (RiboSeq support) values for miSS of different expression categories in protein-coding regions (right). **(C)** The average PhastCons scores (100 vertebrates) for positions near the miSS in different expression categories. **(D)** The distribution of average PhastCons scores (100 vertebrates) at the consensus dinucleotides of splice sites (top) and average PhastCons scores of the adjacent 30 nt intronic regions (bottom). **(E)** The distribution of shifts for non-significant, non-tissue-specific and tissue-specific donor and acceptor miSS.
(TIF)

**S7 Fig. Examples of tissue-specific and non-tissue-specific miSS. (A)** A thyroid-specific miSS in exon 14 of the gene *CAMK2B*. The miSS becomes a maSS in thyroid as its $\varphi$ value

exceeds 0.5. **(B)** The miSS in exon 8 of the *TRABD* gene is non-tissue-specific.
(TIF)

**S8 Fig. NAGNAGs and GYNNGYs. (A)** The intersection of the acceptor miSS located ±3 nts from the maSS with the list of NAGNAGs provided by Bradley et al [5]. **(B)** A NAGNAG acceptor splice site in the exon 20 of the *MYRF* gene. The upstream NAG is upregulated in the stomach, uterus, adipose tissues and downregulated in the brain. **(C)** The intersection of the donor miSS located ±4 nts from maSS with the list of GYNNGYs provided by Wang et al [1]. **(D)** A GYNNGY donor splice site in the exon 2 of the *PAXX* gene. The downstream GY is upregulated in the brain and downregulated in the stomach, pancreas, and liver tissues. **(E)** The response of GYNNGY miSS to NMD inactivation.
(TIF)

**S9 Fig. The response of a miSS to inactivation of an RBP in HepG2 (x-axis) and HepG2 (y-axis) cell lines.** Fractions of significant miSS-RBP pairs located in each quadrant are shown (the fractions are summed to 100%).
(TIF)

**S10 Fig. Structural annotation of miSS. (A)** The comparison of the structural annotation assigned directly to miSS (left) or from the structural annotation of the corresponding maSS (right). Only exonic miSS and corresponding maSS are considered. **(B)** The structural annotation for different categories of miSS.
(TIF)

**S11 Fig. Evolutionary selection of miSS. (A)** The definition of the consensus (Cn) and non-consensus (Nc) nucleotide variants in the donor splice site. The definition for acceptor splice site is similar. **(B)** The evolutionary tree used to reconstruct the ancestral sequence of human and marmoset. **(C)** The computation of *obs* and *exp* statistics. **(D)** The selection of cryptic and not significant miSS in coding regions for marmoset and human genomes. **(E)** The strength of the selection in selected categories of splice sites in the non-coding regions **(F)** The distribution of ancestral strength for different splice site categories. **(G)** The strength of the selection acting on constitutive coding splice sites matched to miSS and maSS by the ancestral splice site strengths. **(H)** Estimation of the 95% confidence interval of $\alpha$ for different expression categories of miSS.
(TIF)

**S12 Fig. The constructed miSS catalogue extends TASSDB2 database. (A)** The intersection of the set of expressed miSS with TASSDB2. **(B)** miSS not contained in TASSDB2 have on average lower $\varphi$ values than miSS in TASSDB2. **(C)** miSS not contained in TASSDB2 are enriched with tissue-specific and non-tissue-specific significantly expressed miSS (top); within these categories they have similar or higher $\varphi$ values compared with miSS in TASSDB2 (bottom).
(TIF)

**S13 Fig. The dependence of the fraction of identified TASS on the number of considered samples.**
(TIF)

**S14 Fig. An example snapshot of the representation of the comprehensive catalogue of human TASS by Genome Browser track hub.**
(TIF)

**S1 Table. Summary statistics at different filtration steps of the TASS catalogue.**
(XLSX)

**S2 Table. Characteristics of miSS in different expression categories.**
(XLSX)

**S3 Table. Abundance of tissue-specific miSS in tissues.**
(XLSX)

**S4 Table. Accession codes for samples of shRNA RNP KD and eCLIP.**
(XLSX)

**S5 Table. miSS-RBP-tissue triples.**
(XLSX)

**S6 Table. Predicted cases of miSS regulation by RBP with eCLIP support.**
(XLSX)

**S7 Table. miSS reactive to PTBP1 KD and OE.**
(XLSX)

**S8 Table. Expressed miSS.**
(TSV)

## Acknowledgments

All authors thank Drs. Sergei Moshkovskii and Mikhail Gorshkov and their research team for insightful discussions.

## Author Contributions

**Conceptualization:** Aleksei Mironov, Olga V. Kalinina, Dmitri D. Pervouchine.

**Data curation:** Aleksei Mironov, Dmitri D. Pervouchine.

**Formal analysis:** Aleksei Mironov, Stepan Denisov.

**Funding acquisition:** Dmitri D. Pervouchine.

**Investigation:** Aleksei Mironov, Stepan Denisov, Olga V. Kalinina, Dmitri D. Pervouchine.

**Methodology:** Aleksei Mironov, Stepan Denisov, Alexander Gress, Olga V. Kalinina.

**Project administration:** Dmitri D. Pervouchine.

**Resources:** Alexander Gress.

**Supervision:** Olga V. Kalinina, Dmitri D. Pervouchine.

**Visualization:** Aleksei Mironov, Stepan Denisov.

**Writing – original draft:** Aleksei Mironov, Dmitri D. Pervouchine.

**Writing – review & editing:** Aleksei Mironov, Olga V. Kalinina, Dmitri D. Pervouchine.

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
