## [Decision Letter · Decision Letter 0]

22 Oct 2020

Dear Prof. Pervouchine,

Thank you very much for submitting your manuscript "An extended catalogue of tandem alternative splice sites in human tissue transcriptomes" for consideration at PLOS Computational Biology.

As with all papers reviewed by the journal, your manuscript was reviewed by members of the editorial board and by several independent reviewers. In light of the reviews (below this email), we would like to invite the resubmission of a significantly-revised version that takes into account the reviewers' comments.

We cannot make any decision about publication until we have seen the revised manuscript and your response to the reviewers' comments. Your revised manuscript is also likely to be sent to reviewers for further evaluation.

Sincerely,

Ilya Ioshikhes

Associate Editor

PLOS Computational Biology

William Noble

Deputy Editor

PLOS Computational Biology

Reviewer's Responses to Questions

**Comments to the Authors:**

Reviewer #1: The study by Mironov, et al, examined the catalog of human tandem alternative splicing sites (TASS) by integrating data from multiple databases, including TASSDB2, GENCODE, UCSC, and Genotype Tissue Expression (GTEx). The authors constructed TASS clusters by grouping alternative splicing sites which are within 30nts, and in each cluster, one major splice site (maSS) was identified and the rest were regarded as minor (miSS). The author further classified the miSS TASS into different categories based on their expression patterns evaulated using GTEx data. They found that miSS with tissue-specific significant expression are conserved as maSS while the rest of miSS are much less conserved and probably full of splicing noise.

In general, the study is well designed and the analyses are reasonable. I have the following comments to

hopefully improve the manuscript.

Major:

1. More explanation is needed for Fig. 3C. How was the number of co triples derived? Is this calculation controlled for the number of tissues in which a splicing event is significantly expressed?

2. In Fig. 4A, it seems that the authors use the group of not-significant miSS as reference, which can be a negative control. Another two refrerences such as consitutive SS and maSS can be positive controls, which represent the distributions for functional SS. Also, what is the "protein" for structure categories?

3. In Fig. S11C, how was the expected value from intronic regions computed? How were the consensus sequences in intronic regions defined?

4. In Fig. 5D, the alphas were estimated using the Cn-to-Nc substitutions? Would the estimates from the Nc-to-Cn substitutions match?

5. In Fig. S11D, why are the selection schemes different between human and marmoset lineages?

6. The supplementary tables are not attached to the pdf manuscript and no link for them.

Minor:

1. provide the link for downloading decoy splice sites from reference 29, or provide more details of how to get the data.

2. authors may consider using a histogram to show the distribution in Fig. 1D, because the CDF plot isn't easy to catch the information.

3. Page 5, line 161, Fig. 7D is referred, but there is no Fig. 7D provided.

4. In Fig. 1E, the 'coding' panel at the left, the scale for the parameter (splice site usage) should be changed to show the difference among groups.

Reviewer #2: Here, Mironov et al present an updated catalog of tandem alternative splicing sites based on analyses of the data from the GTEx resource. The question about whether more than one isoform is used in most tissues is a longstanding one in the splicing field. Tandem alternative splicing sites are prevalent in the human genome and they have been associated with multiple different diseases. Even though they have been characterized previously, large volumes of data have become available over the last few years. Thus, this reviewer believes that this manuscript is a timely contribution as it is important to revisit these types of questions at regular intervals in the light of new data and findings. Overall, the manuscript is well written and the structure is logical and easy to follow. In addition to presenting genome-wide statistics, the authors also highlight multiple specific examples which I find very helpful. Nevertheless, I believe that there are several major issues that need to be addressed:

54: The authors used MaxEntScan to score putative splice sites and to discover potential novel cryptic splice sites that do not exhibit expression on the analysed RNA-seq data. They used a score threshold to only get a confident list of non-expressed cryptic sites, however it is unclear how they determined the optimal threshold.

54: Similarly, Jaganathan et. al. (Cell, 2019) have shown that SpliceAI provides significantly more specificity to predict splice sites from genomic sequence alone. The authors need to address the potentially large fraction of false positive splicing events reported by MaxEntScan by employing an alternative computational strategy, e.g. SpliceAI.

70: The authors say that “almost a half of the expressend splice sites are de novo”. This statement is vague and the authors should provide the exact number or percentage of the total amount of expressed splice sites that were identified de novo.

Moreover, it is unclear if when they say “splice sites” they are referring to the total amount of splice sites or just the ones which support TASS. According to the Method section line 416, only 3 reads from the whole set of RNA-seq data are required to identify a splice site as expressed. Since detection of splice sites using RNA-seq is subjected to mapping errors and technical artefacts during library preparation and sequencing, it is unclear if author statement on line 70 will hold after making sure that the detected splicing events are not false positives. To reduce the number of false positives, authors do not consider novel splicing events that were flanked by annotated polymorphic sites, but this would not account for mapping errors that could be induced at lower frequency alleles. Therefore direct assessment of the mapping quality of the reads that support novel splice junctions might still be required, for example by not considering novel splice junctions that are only evidenced by reads aligned with indels around the detected splice sites. Moreover, authors could also ignore the novel splice sites that are found in only one RNA-seq sample to reduce the number of false splice site detection that is driven by experimental errors and genomic variability.

81: Authors claim to significantly extend the number of TASS that are annotated in TASSDB2, by reporting 32,415 which are not in this database. However, this reviewer is not convinced that a larger number of splice sites is necessarily better. The authors should clarify how many of these TASSs are found when stronger criteria to avoid false positive discovery of novel splice sites are applied.

81: How many of the sites found in TASSDB2 are not included in this study or not detected as expressed in the RNA-seq data?

81: One of the nice features of TASSDB2 is that it has a webserver to host their data. The data presented in this study are not as accessible and the authors should ensure that it is easy for others to access to ensure that this updated catalog is used by other researchers.

146 : What is the φ distribution for the significant and non-significant TASS events?

152 : Authors should consider using a minimum φ value to determine if a TASS is tissue-specific. This can ensure not only a significant deviation, but also a value that could have biological relevance.

154 : Authors report 2,014 tissue-specific miSS. Do any of these miSS become a maSS in any of the tissues analyzed?

154: All differences reported for tissue specific miSS should be supported by numbers or supplementary figures in addition to statistical analyses.

191 : The authors should check if these events are significantly up-regulated as a group after NMD inactivation using the data that they already analyzed in this project.

213: Authors developed a statistical framework to access alternative splicing changes of TASS events after the down-regulation of different protein factors. Given that there are a variety of softwares such as DEXeq, rMATS, and Whippet that can assess alternative splicing changes of TASS as well as other types of alternative splicing events. The main limitation of the computational tools mentioned above is the need for a list of annotated splice events, generally supplied as a GTF file, limiting the analysis to annotated events. However, the authors could generate a new GTF file containing all the new splice sites discovered and use a published software to assess alternative splicing TASS changes. This will ensure the detection of robust alternative splicing changes based on a statistical framework that has been proven to have a good performance handling biological/technical RNA-seq replicates and complex alternative splicing changes. Similarly, changes in gene expression can be assessed with a diverse array of publicly available software and results of differentially expressed genes are available in ENCODE. The authors only mentioned the number of miSS-RBP-tissue triplets that they were able to detect, but they do not mention how many alternative splicing changes or gene expression changes they detected. To validate the computational strategy developed by the authors they should report how many events could be also detected using existing tools.

213: Across this section the authors analyzed a large collection of quantitative measurements derived from RNA-seq and eCLIP data. However, the details of the results are limited. For example, authors found 138 co-directed and 93 anti-directed miSS-RBP-tissue triplets, but they do not provide a figure to allow readers to visualize these changes across this data. Also, when they integrated eCLIP peak results, the authors found 7 miSS-RBP candidate pairs that are supported by eCLIP binding patterns across splice sites. However, it is unclear if the eCLIP peaks distribute differently across TASSs that are predicted to be regulated by these RBPs in comparison with all TASSs analyzed. Finally, figure 3D only shows changes relevant to one of the 7 miSS-RBP candidate pairs supported by eCLIP data across a subset from all the data analyzed. Since these analyses are highly relevant and were performed across a large set of data, authors should provide better ways to visualize the data.

264 : The statement “The residues in this part of the helix become more hydrophobic, which may influence the overall helix or protein stability” is currently not supported by quantitative analyses. Given the array of different available computational tools that can be used to assess protein stability and structure prediction of protein domains, authors should perform a quantitative analysis to suggest this or at least cite relevant literature to backup this claim.

Overall this manuscript lacks substantial biological novelty beyond additional events being detected and identification of possible regulatory RBPs associated to TASSs. To gain further biological insight authors could for example try to further analyze the genomic variants associated to quantitative miSS changes or explore how TASS is regulated through cell-types using publicly available TRAP-seq or scRNA-seq data.

I have also identified the following minor issues:

50: Mln is not a commonly used abbreviation in the literature and it is not currently defined here.

67: Authors should give the exact number of splice sites and express the numbers corresponding to each category as an approximated percentage.

70: In this context an exon could be cataloged as “non-coding” for proteins located at UTRs or genes which do not code for proteins. The numbers would be more clear if authors provide separate numbers for TASS that belong to UTRs or non-coding genes.

78: This is expected, however authors have not explained which expression unit they are using. Does Rn just refer to the number of reads? In the case Rn corresponds to the number of reads, it would be better to use an expression unit that is not confounded by gene expression, such as PSI or the alternative metric the authors introduced.

Table 1: The two bottom columns from `% of split reads supporting TASS` column should add up to 100? The sum of the numbers provided by the authors is 100.1.

85: Authors should carefully check for grammar mistakes, for example here “from the split reads” should just be “from split reads”.

86: “Indeed, we checked that only 2.2% of split reads that support miSS on one end support several splice sites on the other end” is not very understandable. I was expecting you to report the percentage of novel splice junctions that were in your data which “neither donor nor acceptor splice site is annotated”.

124: “..while miSS located upstream tend to be expressed stronger than miSS located downstream “. Is an interesting claim that should be backed up by statistical analyses.

128 : Statistical analyses are missing.

129 : Statistical analyses are missing.

268 : Does an alternative acceptors that are 39 nt apart still count as tandem alternative spice stites? Which is the maximum distance at which alternative splicing 5’/3’ splicing events are considered as TASS?

296 : Authors should report the p-value and statistical test utilized to assess corresponding statistical significance.

309 : The statement “we observed only a subtle difference in evolutionary selection between” it is vague. Authors should report the magnitude of the difference and some parameters to claim these are just subtle differences.

Figure 1E: Colours need to be explained.

Figure 1F: The significance should be coded by *, **, *** marks. The exact p-value and statistical test should be included in the figure legends or main text.

Figure 2E: This figure should be wider.

Figure 4A-D and Figure 5A-B: Authors should explain the meaning of the error bars and highlight any statistical difference found while comparing these measurements.

Reviewer #3: Mironov et al. present a new, more comprehensive catalogue of human TASS cases thas has been compiled based on recent RNA-seq data. In my view, this alone has only minor impact for the research field. More interesting is the investigation on tissue specificity of TASS isoform ratios. However, presentation of this latter part is quite condensed and difficult to track. The manuscript should be improved to clarify the specific outcome of the analysis and correctly put it into context of the heterogenous TASS catalogue. This is particularly important for the proposed mechanism of PTB acting as a tissue-specific regulator of TASS isoform formation.

MAJOR

1. The authors claim that they "substantially extend the existing catalogue of TASS" (l. 37), which is probably correct. The significance of this progress should be analyzed with respect to significance of TASS outcome. TASS isoform products are the more likely to be functionally relevant the more balanced the isoform ratios (high phi values) are. One can speculate that TASS cases identified in this study are the ones with very low miSS (less likely to be functionally relevant) because this would be an explanation why previous studies (using less sequence data) have overlooked these cases. The authors should analyze the phi value in the newly identified TASS cases in relation to previously known cases.

2. Important previous studies on tissue-specific TASS are not cited and not discussed: DOIs 10.1101/gr.186783.114 and 10.1093/nar/gku532, as far as i oversee. These must be included in a general outline on models of TASS regulation.

3. When it comes to functional characteristics, esp. tissue-specificity of splicing, Mironov et al. hardly differentiate the TASS subtypes. Only the NAGNAG subtype (acceptors in 3 nt distance), probably the largest subgroup, is analyzed separately. 429 of 7414 NAGNAG cases (5.8%) appear to be less frequently tissue-specific compared to TASS average, 2014 of 12361 (16.3%). Tandem donors form another specific subtype, which deserve specific consideration. A separation of the subtypes offer important mechanistic insights. Splice site distance is another relevant structural property - see next point.

4. How general is the proposed mechanism of PTBP1 acting as a tissue-specific regulator of TASS isoform formation? This is an important question. I suppose, and this should definitely be tested, that a regulatory involvement of PTB in tissue-specific splicing is positively associated with splice site distance. This is likely because PTB binds to the polypyrimidine tract; a polypyrimidine tract overlapping the TASS region, the longer the more efficient, would be a plausible action platform for PTB interference.

5. What is the fraction of RBP association with tissue-specific TASS that is explained by PTB (or other particular factor)? The fraction of PTBP1-associated tissue-specific TASS in total? This would hint to unexplored contributors of tissue-specificity. The supplement announces table data (I cannot inspect) towards this question but, anyway, these general questions need to be addressed in the main text.

6. Fig. 2, panel A is meant to illustrate evidence for alternative splicing of TASS cases. However, highlighting of NPTN (tissue-specific TASS example) suggests it might illustrate tissue-specific TASS splicing. To make the steps fully clear, I suggest to place panel B as panel A; panel A as panel B omitting the NPTN highlight and add an additional panel (neo)C which shows the separation of tissue-specific and non-tissue-specific TASS cases with the NPTN highlight. QKI, the tissue-specific example illustrated in fig. 3, must also be highlighted.

7. In the methods to detect regulation of miSS by RBP (l. 476 ff), how is the background distribution of the slope modeled? This should be specified. Same for tissue-specific miSS (l. 456 ff).

MINOR

8. Reference to cystic fibrosis as a severe disease caused by single-aa indel [11] (l. 15) in the context of TASS is misleading because the variant is a mutation, which is subject to purifying selection (although balaced by minor advantageous effects). In contrast, TASS generates isoform molecules from the same allele, likely have passed purifying selection (esp. with equi-expressed isoforms), may be even subject to positive selection (gain of function). This reference should be omitted or made clear by explanation.

9. The statistics for miSS expression has a flaw in correcting for multiple testing. As the authors state (l. 449), multiple testing is corrected by a Q-value metric at the level of individual tissue. However, in the analysis of multiple tissues the Q-value metric is no longer valid to describe the meta-significance appropriately. The metric needs to be adjusted to nested multiple testing.

10. What is the straight line Fig. 2A representing? Apparently, it is not relevant for separation of significant and non-significant miSS (tracability of minor isoform).

**Have all data underlying the figures and results presented in the manuscript been provided?**

Reviewer #1: Yes

Reviewer #2: **No: **The supplementary material states, "ATTACHE XLS FILE" referring to tables S1-6, but no such file could be accessed. Also, from the description it does not seem as if there is a table containing details of all miSS and maSS analyzed which is something that would be required for reproducibility.

Reviewer #3: Yes

PLOS authors have the option to publish the peer review history of their article (what does this mean?). If published, this will include your full peer review and any attached files.

Reviewer #1: **Yes: **Zhenguo Zhang

Reviewer #2: No

Reviewer #3: No
---

## [Decision Letter · Decision Letter 1]

14 Feb 2021

Dear Prof. Pervouchine,

Thank you very much for submitting your manuscript "An extended catalogue of tandem alternative splice sites in human tissue transcriptomes" for consideration at PLOS Computational Biology. As with all papers reviewed by the journal, your manuscript was reviewed by members of the editorial board and by several independent reviewers. The reviewers appreciated the attention to an important topic. Based on the reviews, we are likely to accept this manuscript for publication, providing that you modify the manuscript according to the review recommendations.

Sincerely,

Ilya Ioshikhes

Deputy Editor

PLOS Computational Biology

William Noble

Deputy Editor

PLOS Computational Biology

[LINK]

Reviewer's Responses to Questions

**Comments to the Authors:**

Reviewer #1: By incorporating the comments from the reviewers, the manuscript has been

improved in clarity and reliability. The authors have addressed most of my

concerns, and here are a few remaining things needing authors' inputs:

1. In Fig 5A (4A in old version), I proposed to show the structure category

distribution for maSS and constitutive splicing sites, because they represent

the distributions for functional splicing sites. Will the significant expressed

tissue-specific miSS (supposed to be functional) have a similar distribution with those distributions?

2. Page 3, Line 56, it states that there are ~600K cryptic sites, but Table S1A

shows only 196885 sites. Please explain it.

Reviewer #2: Overall I think the authors have improved the manuscript significantly and they have addressed all of my comments. Nevertheless, there are some minor issues remaining.

1) they said that in the original submission they took care of reads aligning with miss matches around splice sites, but they just mentioned filtering steps related with annotated polymorphisms.

2) I agree with the authors that making the data available as UCSC hub, however neither within the tracks or in the manuscript there I could find documentation that enables me to fully interpret the data displayed on these tracks (e.g. explanation of the different colors).

3) even though it might not be author’s fault, the figure quality on this submission was very low, making it quite hard to interpret the results.

Reviewer #3: All my remarks have been addressed properly.

**Have all data underlying the figures and results presented in the manuscript been provided?**

Reviewer #1: Yes

Reviewer #2: Yes

Reviewer #3: Yes

PLOS authors have the option to publish the peer review history of their article (what does this mean?). If published, this will include your full peer review and any attached files.

Reviewer #1: No

Reviewer #2: No

Reviewer #3: No
---

## [Decision Letter · Decision Letter 2]

22 Mar 2021

Dear Prof. Pervouchine,

We are pleased to inform you that your manuscript 'An extended catalogue of tandem alternative splice sites in human tissue transcriptomes' has been provisionally accepted for publication in PLOS Computational Biology.

Best regards,

Ilya Ioshikhes

Deputy Editor

PLOS Computational Biology

William Noble

Deputy Editor

PLOS Computational Biology

Reviewer's Responses to Questions

**Comments to the Authors:**

Reviewer #1: All my concerns have been well resolved. Thanks.

**Have all data underlying the figures and results presented in the manuscript been provided?**

Reviewer #1: Yes

PLOS authors have the option to publish the peer review history of their article (what does this mean?). If published, this will include your full peer review and any attached files.

Reviewer #1: No

---

## [Editor Report · Acceptance letter]

4 Apr 2021

PCOMPBIOL-D-20-01627R2 

An extended catalogue of tandem alternative splice sites in human tissue transcriptomes

Dear Dr Pervouchine,

I am pleased to inform you that your manuscript has been formally accepted for publication in PLOS Computational Biology. Your manuscript is now with our production department and you will be notified of the publication date in due course.

With kind regards,

Alice Ellingham
